# On Low Frequencies Fourier Features in Reinforcement Learning

## Abstract

Recent reinforcement learning (RL) methods have adopted ideas from image processing tasks by employing Fourier Features (FFs) encoding. This approach enables a typical multilayer perceptron (MLP) to learn different frequency features. However, a disparity exists between the scale of frequencies used for image and RL problems. Previous works employed significant lower frequencies to successfully train RL agents and defer to the Neural Tangent Kernels (NTK) theory for justification. However, we observed that NTK cannot provide satisfactory explanations. We present a novel perspective empirically to show why lower frequencies are essential for the successful training of RL agents. Our empirical investigation is based on the cross-correlation among state dimensions and their overall cross energy spectral density (CSD). Based on our empirical observation, we propose a simple enhancement to the current FFs formulation and achieve performance improvements over current FFs formulation and baseline methods.

## 1 Introduction

It has been shown in prior research (Rahaman et al., 2019; Cao et al., 2021; Fridovich-Keil et al., 2022) that during the training process, neural networks tend to prioritise the extraction of low-frequency information over high-frequency details. In other words, they initially capture broad global patterns within the data before progressively refining their understanding to encompass finer-grained, higher-frequency components. This newfound theory of neural networks training, known as the spectral bias (Rahaman et al., 2019), has proven applicable to techniques such as positional encoding within neural radiance fields (NeRFs) (Mildenhall et al., 2020). This enables neural networks to directly learn high-frequency details, which turns out is a critical capability for generative tasks involving image-based data.

Similar concept has also been explored in reinforcement learning (RL). Recent investigations (Konidaris et al., 2011; Li & Pathak, 2021; Yang et al., 2022; Brellmann et al., 2023) corroborates the potential for manipulating frequencies to learn robust policies and state-action value functions. Although empirical evidence suggests that using high frequencies in Fourier Features (FFs) is essential for optimal performance in NeRFs (Mildenhall et al., 2020) and related implicit neural representation (INR) tasks (Benbarka et al., 2021; Tancik et al., 2020), recent research contradicts this notion in the contexts of RL. Employing significantly lower frequencies in the FFs is essential for robust RL learning (Li & Pathak, 2021; Yang et al., 2022). However, existing literature primarily focuses on justifying the empirical effects of employing FFs for RL using the theory of Neural Tangent Kernels (NTK) (Jacot et al., 2018).

The motivation for this work stems from our observation that, while lower frequencies were employed in previous works employing FFs for RL (Li & Pathak, 2021; Yang et al., 2022), their use of NTK theory cannot comprehensively justify this choice of lower frequencies. This observation prompts us to investigate the change in behaviour of the state information when encoded with FFs using lower and higher frequencies.

In summary, the contributions of this work are as follows:

- We show empirically that NTK theory, a tool that previous works use, cannot comprehensively explain why lower frequencies in FFs are essential in the contexts of RL.

- We present an alternative perspective to understand the significance of using lower frequencies in FFs, which empirically are shown to be essential in the contexts of RL. Our analysis of encoded state information is based on observations of cross-correlation among state dimensions and consequent energy distribution among them when FFs are employed for RL. We observe certain unique cross-correlation and energy distribution phenomena that emerge exclusively when lower frequencies are employed. We hypothesise that augmenting with FFs effectively changes the energy distribution, thus the importance of certain features in the state information.

- Consequently, we propose a simple enhancement to complement the current FFs method for RL. While recent FF-based methods use frequencies that follow a zero-mean Gaussian distribution, we demonstrate that this approach is not necessarily optimal. We propose an alternative approach to choosing frequencies based on a fixed spacing scheme, which can lead to performance gains over current FFs and baseline methods.

- We further verify that our proposed enhancement are less sensitive to hyperparameter choice regarding FFs, and can be justified based on our alternative perspective and empirically.

## 2 Preliminaries

### 2.1 Reinforcement Learning

A standard RL problem is defined as an infinite-horizon Markov Decision Process (MDP) $= \langle \mathcal{S}, \mathcal{A}, p, \mathcal{R}, \gamma \rangle$, where the RL agent at timestep $k$ observes a state $s_k$ from a set of states $\mathcal{S}$, chooses an action $a$ from a set of actions $\mathcal{A}$, and receives a reward $r$ according to a mapping of the reward function $\mathcal{R}$, $r : \mathcal{S} \times \mathcal{A} \to \mathbb{R}$. The environment then transitions into a state $s_k'$ with a transition probability function $p$ and the interaction continues. We define the replay buffer $\mathcal{D}$ containing a set of state, action, reward, and next state at timestep $k$ as $\mathcal{D} = \{(s_k, a_k, r_k, s_k')\}_{k=1}^{N}$ with $N$ denoting a replay buffer size hyperparameter. The objective of an RL agent is to find a policy $\pi$ that maximise the discounted expected return $\mathbb{E}_\pi[\sum_{t=0}^{\infty} \gamma_k R(s_k, a_k)]$, which is the expected cumulative sum of rewards when following the policy in the MDP, where the importance of the horizon is determined by a discount factor $\gamma \in [0, 1)$.

### 2.2 Temporally consecutive states

Each state $s_t$ at time $t$ may contain $j$ state dimensions, $\{s_{t_k}, s_{t_l} ..., s_{t_j}\}$ where $\{k, l \in \mathbb{Z} | 0 \leq k < l \leq j\}$. As illustrated in Figure 1, for instance, a state of a robot can contain several joints, such as certain joints in robot-leg one and certain joints in robot-leg two. In the analysis in Section 4 of this work, we use states that are sampled from the replay buffer $\mathcal{D}$ using consecutive indices, so that consecutive states from the same trajectory were used in the analysis and temporal continuity is explicitly conserved.

Figure 1: Example illustrating the state dimensions. We calculate the cross-correlation for each state dimension pairs in Section 4.

### 2.3 Cross-correlation and Autocorrelation

With $t$ denoting the time, the cross-correlation describes the similarity between two time-dependent discrete timeseries $x[t]$ and $y[t]$ that are shifted against each other on the time axis, given a time lag $\tau$ between the points. It can be used to identify overlapping periodic components in the timeseries $x[t]$ and $y[t]$. The Autocorrelation is a special case of cross-correlation where $x[t] = y[t]$ and is a tool for identifying recurring patterns in a timeseries $x[t]$ and distinguishing periodic components from noise. Specifically, we calculate the cross-correlation between two state dimensions pairs $\{s_{t_k}, s_{t_l}\}$. Some toy example can be find in Section A.

**Definition 2.1** (Cross-correlation). *The cross-correlation of two discrete time-dependent timeseries $x[t]$ and $y[t]$ is defined as:*

$$R_{xy}[\tau] := \sum_{t=-\infty}^{\infty} x[t]y[t+\tau] \tag{1}$$

**Definition 2.2** (Autocorrelation). *The autocorrelation of a discrete timeseries $x[t]$ is defined as:*

$$R_{xx}[\tau] := \sum_{t=-\infty}^{\infty} x[t]x[t+\tau] \tag{2}$$

### 2.4 Energy Spectral Density and Cross Energy Spectral Density

The cross energy spectral density (CSD) quantifies the relative energy distribution across frequencies for the cross-correlation function, $R_{xy}$, between two timeseries, $x[t]$ and $y[t]$. The CSD at a specific frequency reveals how the energy of $x[t]$ and $y[t]$ is jointly distributed across the frequency spectrum (i.e. at which frequency the two signals share more energy). It is also the Fourier transform of $R_{xy}$. Conversely, the energy spectral density (ESD) defines the energy distribution of individual frequency components within a single timeseries. It highlights the frequencies where the timeseries exhibits significant or minimal energy.

**Definition 2.3** (Cross Energy Spectral Density). *Given two finite timeseries $x[t]$ and $y[t]$, the cross energy spectral density (CSD) is defined as:*

$$S_{xy}(f) := \sum_{\tau=-\infty}^{\infty} R_{xy}[\tau]e^{-i2\pi f\tau} \qquad S_{yx}(f) := \sum_{\tau=-\infty}^{\infty} R_{yx}[\tau]e^{-i2\pi f\tau} \tag{3}$$

**Definition 2.4** (Energy Spectral Density). *Given a timeseries $x[t]$, the energy spectral density (ESD) is defined as:*

$$S_{xx}(f) := |\hat{x}[f]|^2, \ where \ \hat{x}[f] = \sum_{t=-\infty}^{\infty} x[t]e^{-i2\pi ft} \tag{4}$$

### 2.5 Kernels and Random Fourier Features

In the context of kernel methods, Rahimi & Recht (2007)'s seminal work on RFFs presented them as a means to approximate a given kernel $K$. While cosine is used in the seminal formulation, subsequent investigations (Sutherland & Schneider, 2015) changed RFFs with using sine FFs.

**Definition 2.5** (Random Fourier Features (Rahimi & Recht, 2007)). *With frequency $f_i \sim \mathcal{N}(0, \sigma^2)$ for $i = 0, \ldots, n$ and $b \sim \mathcal{U}(0, 2\pi)$, the Random Fourier Features are derived as*

$$K(x - x') = \mathbb{E}[2\cos(fx + b)\cos(fx' + b)]$$

$$\approx \frac{1}{n} \sum_{i}^{n} \sqrt{2}\cos(f_i x + b_i)\sqrt{2}\cos(f_i x' + b_i) = \boldsymbol{Z}(x)^{\mathsf{T}}\boldsymbol{Z}(x') \tag{5}$$

$$where \ \boldsymbol{Z}(x) = [\sqrt{\frac{2}{n}}\cos(f_1 x + b_1), \sqrt{\frac{2}{n}}\cos(f_2 x + b_2), ..., \sqrt{\frac{2}{n}}\cos(f_n x + b_n)]^{\mathsf{T}}$$

$$\text{is the Random Fourier Features (RFFs).} \tag{6}$$

While different formulations exist, previous works (Mehrkanoon & Suykens, 2018; Tancik et al., 2020; Li & Pathak, 2021; Yang et al., 2022) simplified the original formulation as the following. Unless specified otherwise, this is the definition we will use throughout this work, while we briefly discuss the effect of different formulations in Section C.

$$\text{RFFs}(x) = [\cos(2\pi f_1 x), \sin(2\pi f_1 x), ..., \cos(2\pi f_n x), \sin(2\pi f_n x)]^{\mathsf{T}} \tag{7}$$

$$\text{where frequencies } f_i \sim \mathcal{N}(0, \sigma^2) \text{ for } i = 0, \ldots, n$$

### 2.6 The Neural Tangent Kernel and its Relationship to Fourier Features

The Neural Tangent Kernel (NTK) is a kernel function that describes the evolution of neural networks during training.

**Definition 2.6** (Neural Tangent Kernel). *Let $f(x; \boldsymbol{\theta})$ denote the scalar function computed by a given neural network with parameters $\boldsymbol{\theta}$ on input $x$. Assuming the network has infinite width and is trained via stochastic gradient descent, the Neural Tangent Kernel (Jacot et al., 2018) is defined as:*

$$k(x_i, x_j; \boldsymbol{\theta}) = \langle \nabla_{\boldsymbol{\theta}} f(x_i; \boldsymbol{\theta}), \nabla_{\boldsymbol{\theta}} f(x_j; \boldsymbol{\theta}) \rangle \tag{8}$$

The convergence rate of a neural network during training is closely related to the eigenvalues of the Neural Tangent Kernel (NTK) (Arora et al., 2019; Yang & Salman, 2020; Basri et al., 2020). For a conventional MLP, the eigenvalues of the NTK decay rapidly, due to the resulting narrower kernel width.

In the seminal work of Tancik et al. (2020), FFs were shown to be able to tune the kernel width of the NTK function, therefore affecting its resulting eigenvalues. By doing so, it is possible to tune to which frequencies the neural network should focus on, mitigating the spectral bias of neural networks and improve their ability to learn high-frequency features.

## 3 Motivation: Gap Between NTK-Based Theoretical Analysis and Practical Implementation

Building on the established analyses in the existing literature (Li & Pathak, 2021; Yang et al., 2022), following (Tancik et al., 2020), we first plot the NTK function of a three-layer MLP network, given inputs from $[0, 1)$, when employing FFs for RL tasks with lower frequencies $f$ sampled from various $\sigma$, $f \sim N(0, \sigma^2)$.

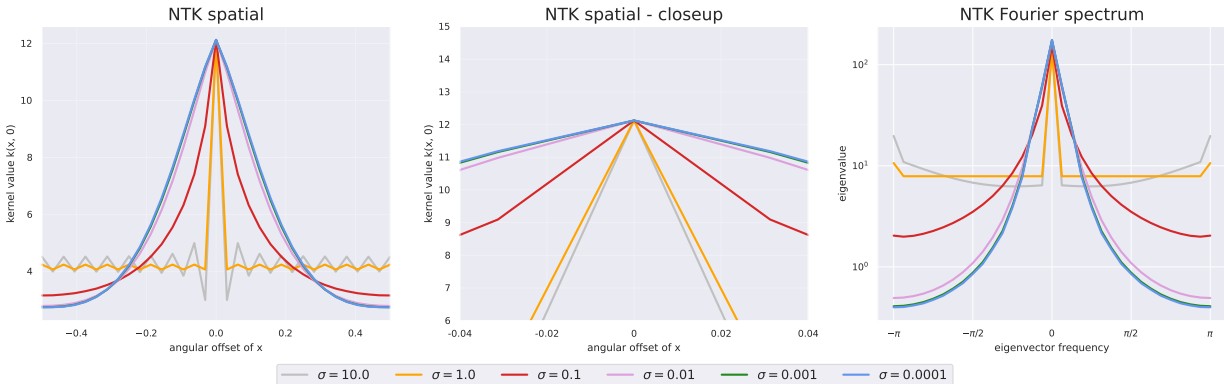

Figure 2: NTK function for various FFs $f \sim N(0, \sigma^2)$. The left plot shows the spatial NTK. The middle is an enlarged version of the left. The right plot shows the spectral plot of NTK after Fourier transform. In all three plots, the difference between $\sigma = 1e-05$ (blue) and $\sigma = 0.001$ (green) is almost indistinguishable. An explanation in using lower frequencies for employing FFs for RL is lacking.

We first examine the NTK. The left and middle plot shows the kernel values between distances between points. The further the distance, the smaller its kernel value relative to zero. For the spectrum plot on the right, the x-axis indicates the frequencies and the y-axis shows the corresponding eigenvalues. Previous analyses Tancik et al. (2020) emphasise that because of a larger $\sigma$, the spectral decay in Figure 2 would be slower in higher frequencies, thus allowing faster convergence for high-frequency details. At low frequencies (i.e. around zero in x-axis), only the lower frequencies have large eigenvalues, so they are the only ones first and foremost learnt through training. Increasing $\sigma$ also increases the higher frequencies' eigenvalues, thus enabling them to learn these higher frequency characteristics faster.

While previous works (Li & Pathak, 2021; Yang et al., 2022) point to NTK as a justification for using lower frequencies, if we closely examine the NTK in Figure 2, there is no real difference in terms of the narrowness or width of the kernel for much lower frequencies, therefore there should not be a huge performance difference when training RL tasks using these lower frequencies. Nevertheless, previous works were employing frequencies sampled from *as small as* $\sigma = 0.00005$ to train RL agents (depending on the task), because a larger $\sigma$, such as $\sigma = 1$, would cause the algorithms to overwhelmingly underperform (Yang et al., 2022). It is worth noting that in NeRF (Mildenhall et al., 2020) tasks, $\sigma$ on the scale of tens is often necessary.

This is somewhat counter-intuitive because in practice, much lower frequencies are used but their NTK does not show significant difference. Thus, we argue that there is an empirical disparity in using the NTK theory to justify employing lower frequencies, which were necessary for achieving competent RL performance, as shown in previous works (Li & Pathak, 2021; Yang et al., 2022). We contend that additional, unobserved but influential factors exist.

## 4 Fourier Features Augment the Cross-correlation among State Dimensions

In this work, we aim to provide an alternative perspective on why lower frequencies in FFs are empirically shown to be essential for RL. Although correlation is explicitly disregarded when we randomly sampled from $\mathcal{D}$ to form the mini-batch $\mathcal{D}_{\text{batch}}$ during training, one noteworthy question that we can ask is to understand how the state input $\mathcal{D}_{\text{batch}}$ is affect by encoding with FFs. Naturally we can look at cross-correlations as a tool for analysis, because of the potential for cross-correlation among states after encoding with FFs,

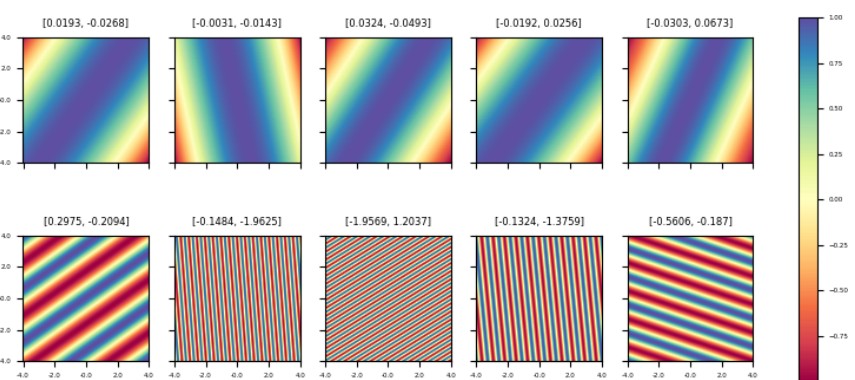

Figure 3: An example of encoding toy input values with FFs over two frequencies $f_1$ and $f_2$. The subtitle is the randomly sampled frequencies used. The x-axis and the y-axis denote toy input values, ranging from -4 to 4. **Top**: lower $f_i \sim \mathcal{N}(0, 0.05)$. **Bottom**: higher $f_i \sim \mathcal{N}(0, 1)$.

which has periodic functions, and the filtering effect of cosine and sine functions which suppresses large change in values into the range of $(-1, 1)$.

To give an intuition of how FFs encoding can effect values, we begin by presenting an example in 2D in Figure 3 to illustrate how FFs encoding affects the input when two lower and two higher frequencies $\omega_1$ and $\omega_2$ are used. When lower frequencies are used, naturally, after encoding, the frequency of change is much smaller, while higher frequencies would represent recurring changes. For lower $f$, the dominant output becomes close to magnitude equals 1 (colour blue) where changes occur slowly. On the contrary, higher $f$ would result in much higher periodic values. Effectively, these toy values are smoothed when lower frequencies are used; only high toy values are preserved. That is, the frequency range that is empirically shown to be essential in RL effectively encodes the state features to similar values, meaning that most states become correlated and only sufficiently abrupt changes in state features remain distinguishable.

To further demonstrate the effect, we start by examining the changes done to a $\mathcal{D}_{\text{batch}}$ where states are sampled from the replay buffer using consecutive indices and examine the cross-correlation among state dimensions before and after FFs encoding. Using Figure 1 as an illustrative example, some joints (i.e. state dimensions) are naturally correlated to other joints, while being negatively correlated or uncorrelated to others. For instance, the angle of a front tip might have a certain correlation with the angular velocity of a front tip, as they move together generally. We can naturally hypothesise that neural networks learn these cross-correlations implicitly.

We demonstrate in this section that FFs offer a method to *explicitly encode cross-correlations*, and different frequencies in FFs can *shift the cross energy spectral density* among state dimensions $\mathcal{D}_{\text{batch}}$. We then

subsequently demonstrate that random sampling frequencies (i.e. RFFs), as depicted in (Rahimi & Recht, 2007) and directly adopted in previous related works on RL (Li & Pathak, 2021; Yang et al., 2022), might not be as beneficial as previously suggested, and we propose an enhancement as a remedy. For the following illustrations, unless otherwise specified, we plot using the task Ant-v5 (Towers et al., 2024).

## 4.1 Cross-Correlation Analysis of Fourier Features Indicates the Properties of Low-Frequency Components

To examine the impact of FFs encodings across varying frequencies, we examine the cross-correlation between two distinct state dimensions in batch $\mathcal{D}_{\text{batch}}$ both before and after encodings. Figure 4 provides a numerical illustration of this cross-correlation. Specifically, we plot the cross-correlation between the state dimensions $s_{t_3}$ (y-orientation of the torso (centre)) and $s_{t_5}$ (angle between torso and first link on front left), using the batch index as a proxy for the time index. The states are sampled from the replay buffer using consecutive indices, so that temporally consecutive states from the same trajectory were used. Further details about the illustration can be found in Section A.

Our analysis of the illustrations reveals two key findings. Originally, prior to encoding $\mathcal{D}_{\text{batch}}$ with the RFFs, both timeseries exhibit high-frequency variations and they were weakly correlated (top row). Intuitively speaking, noisy cross-correlation suggests an absence of discernible cross-correlation. On the other hand, a perfect cross-correlation between two identical timeseries would manifest as a distinct, clean pyramidal pattern.

Firstly, encoding with FFs explicitly introduces cross-correlation *only if we use the lower frequencies* (second to fourth row). This is due to lower frequencies can filter out certain parts of state information, while higher frequencies tend to preserve all information, and even overestimates the importance of certain information. Interestingly, if significant low frequencies are used, the respective cross-correlation becomes closer to a pyramid-like shape, which indicates that the two timeseries become almost identical. This phenomenon diminishes progressively as frequency hyperparameter increase (forth row) and eventually vanishes entirely at higher frequencies (fifth row to last row), and states remain weakly correlated or noisily uncorrelated.

Thus, we hypothesise that only employing FFs with lower frequencies can explicitly filter out certain state information, encode certain cross-correlations and shift the overall energy towards the low-frequency components within the batch $\mathcal{D}_{\text{batch}}$. Our results indicates that employing FFs with lower frequencies can encode meaningful information of state and dynamics. Note that this kind of cross-correlation shape also happens to different pairs of state dimensions, although to various different degrees.

Similarly, the CSD (in the right-most column), which analysis how energy is distributed across frequency spectrum of the states, show that much more energy are concentrated in the low frequency spectrum when encoded with lower frequencies $f$, whereas the energy remains evenly distributed across the high frequency spectrum if we use higher frequencies.

As evidenced by the peaks in the leftmost two columns, $s_{t_3}$ and $s_{t_5}$, since there no cross-correlation before encoding, the energy of these two timeseries is spread more evenly across the spectrum. However, after encoding with low-frequency FFs, higher amplitude dynamics change occur much less frequently. That is, certain originally low-energy features are filtered out by FFs while only high-energy dynamics remain. On the other hand, if we employ high frequency, we might over-energises some dynamics then it should have been.

Plots of different pair of state dimension timeseries can be found in Section E, where we select two one pair where pre-encoded states show some cross-correlation and another pair where the noisy cross-correlation exist. Both pair confirms our finding that low-frequency FFs explicitly filter out certain state information, shift the overall energy towards the low-frequency components within the timeseries. Such effect cannot be readily observed if high frequencies are employed in FFs.

## 4.2 Our Proposed Method: Frequency Sampling Need Not Be Gaussian

We have observed that FFs augment cross-correlation, only if lower frequencies are used. However, how to generate these frequencies can be improved, as long as they augment cross-correlation and shift energy towards low frequency spectrum in the CSD plot. Building on the considerations presented, we propose a simple enhancement to the original RFFs formulation, based on our empirical observations. Specifically, we propose to 1) generate FF frequencies based on a fixed spacing scheme 2) no negative frequencies.

Intuitively, since we cannot determine in advance which feature is more important (i.e. which feature should be given more energy), an intuitive approach is to assign the same feature importance, allowing neural networks to decide which encoded state dimensions hold greater significance as training progresses. A Gaussian distribution, in addition to negative frequencies, is expected to provide more energy to certain state dimensions initially, which may not be necessarily optimal.

We propose to assign $f$ as uniformly spaced frequencies once at initialisation for RL tasks, with each interval calculated by a hyperparameter $\psi$, compared to the original RFFs formulation where $f \sim N(0, \sigma^2)$. The output matrix size of $f$ is determined by the hyperparameter $n$ (ff dimension, which is shared with original RFFs formulation) and $\psi$ determines the maximum frequency (ff scale). We refer to this approach to

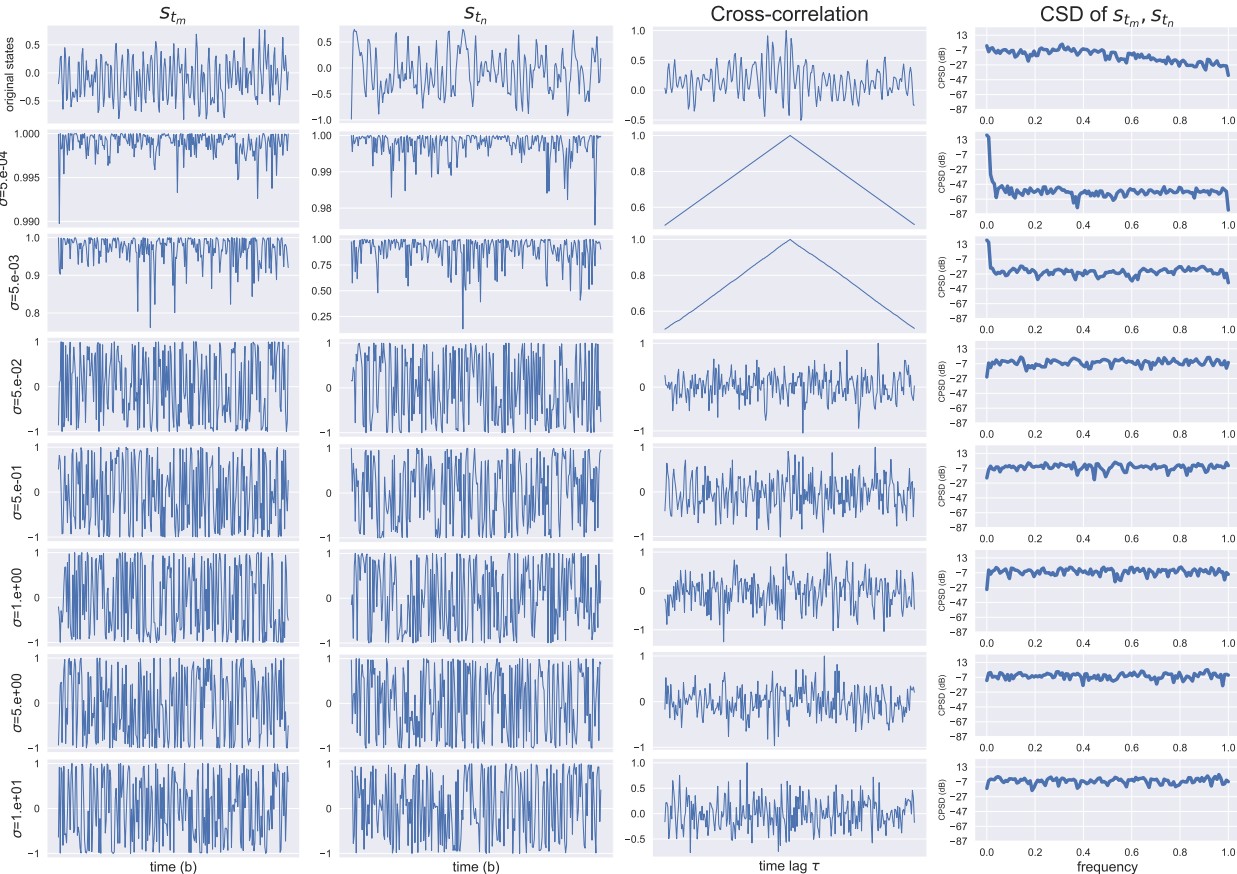

Figure 4: An example illustration of encoding a pair of state dimensions $\mathcal{D}_{\text{batch}}$ with RFFs. The top row denotes pre-encoded state dimensions. The second row to the last row denotes encoded states with frequencies $f \sim \mathcal{N}(0, \sigma^2)$ sampled using various $\sigma$. Originally, both timeseries have higher frequency of change and they were noisily correlated (in the third column). The cross-correlation and CSD become more apparent for smaller $\sigma$ (second row and third row) but remain noisy for larger $\sigma$ (fourth to bottom row). We use the same y-axis scale for the CSD plot.

generating the frequencies as the linear Fourier Features (LinFFs), which is defined as

$$\text{LinFFs}(x) = [\cos(2\pi f_1 x), \sin(2\pi f_1 x), ..., \cos(2\pi f_n x), \sin(2\pi f_n x)]^{\mathsf{T}} \tag{9}$$
$$\text{where } f_i = {}^{i\psi}/n \text{ for } i = 0, \ldots, n$$

A numerical example containing the states, cross-correlation and CSD of our proposed LinFFs in shown in Section B. We generally observe that our proposed LinFFs have have some cross-correlation among state features even with higher frequency. Additional plots for both RFFs and LinFFs of different pairs of $k, l$ state dimensions $\{s_{t_k}, s_{t_l}\}$ where one where some cross-correlation exists before encoding and another where correlation does not exist in Section E.

## 5 Experiments

The main goals of the experiments are (1) to empirically demonstrate that a random sampling scheme is not essential, and that a more suitable frequency sampling scheme can often outperform randomly sampled frequencies, and (2) to illustrate that energy distribution concentrated in the low frequency spectrum is essential for desirable results. Vice-versa, a spread of energy across the entire spectrum, resulted from sampling higher frequencies, does not yield desirable results. It is worth noting that while for the analysis in Section 4 we have explicitly used temporally consecutive states, during experiments below we used the sample randomly indices to gather $D_{\text{batch}}$ from the replay buffer following conventional practices.

### 5.1 Experiment Setups

For all experiments, frequencies , following previous works (Li & Pathak, 2021; Yang et al., 2022).

We conducted our experiments based on two commonly used benchmarking state-based RL algorithms: TD3 (Fujimoto et al., 2018) and SAC (Haarnoja et al., 2018). We selected 5 commonly used tasks from gymnasium (Towers et al., 2024) and 2 tasks from panda-gym (Gallouédec et al., 2021)) to verify our proposal. Gymnasium are locomotion tasks and panda-gym are manipulation tasks. For all experiments, we follow previous works employing FFs in the context of RL (Li & Pathak, 2021; Yang et al., 2022; Brellmann et al., 2023) where frequencies are initialised once using Equation (9) or Equation (7) at the start of training and make the frequencies $f$ trainable. These learned Fourier Features are termed as Fourier Feature Networks (FFN) by (Yang et al., 2022), where they build a layer of learned Fourier Features on top of SAC. We will thus refer to their baseline method that uses RFFs at initialisation as GaussFFN and our proposed LinFFs initialisation as LinFFN. All tasks are trained for $3 * 10^6$ steps evaluated on 30 episodes over 10 seeds, seed $\in [0, 1, 2, 3, 4, 5, 6, 7, 8, 9]$. We aggregate using mean and standard deviation and count the number of first places. If two performances are statistically the same, we count both as first. It is worth noting that compared to previous works employing FFs for RL, we do not introduce more hyperparameters. We vary the frequency hyperparameters $\sigma$ of GaussFFN and $\psi$ of LinFFN per task. Further details on training and the hyperparameters can be found in Section D.

### 5.2 Results on baselines

The results with TD3 as base algorithm are shown in Table 1. We found that most tasks trained with our proposed LinFFN encoding achieved better performance than those initialised with GaussFFN encoding, indicating that the designed frequencies $f$ can outperform previous state-of-the-art methods in employing FFs. The performance gains for SAC in Table 2 are not as apparent, but we generally observe that our proposed LinFFN can generally obtain the best or near-best performance.

### 5.3 Our Proposed Enhancement is Less Sensitive to Frequencies Choice

To further demonstrate the sensitivity to the range of frequencies empirically, we performed experiments with TD3 as base algorithm on two tasks with high frequencies parameter $\sigma$ and $\psi \in \{1.0, 5.0, 10.0\}$. Based on the performance shown in Table 3, depending on the task, we observed that RFFs encoding severely

Table 1: Performance on the gymnasium and panda-gym benchmark with TD3 as base algorithm at 1M and 3M timesteps, calculated over 10 seeds, with ± denoting standard deviation. The highlighted performance is highlighted in blue. Any performance which is not statistically significantly worse than the highest performance (according to a Welch's t-test with significance level 0.05) is also highlighted in yellow.

| Task | Timestep | TD3 | TD3+GaussFFN | TD3+LinFFN |
|------|----------|-----|--------------|------------|
| Ant-v5 | 1M | 3735 ± 986 | 3792 ± 810 | 4950 ± 370 |
| Ant-v5 | 3M | 3919 ± 1049 | 4564 ± 624 | 6260 ± 190 |
| HalfCheetah-v5 | 1M | 9765 ± 1090 | 10847 ± 248 | 10856 ± 760 |
| HalfCheetah-v5 | 3M | 12483 ± 1234 | 12316 ± 565 | 14257 ± 1018 |
| Hopper-v5 | 1M | 3063 ± 701 | 3098 ± 362 | 3296 ± 171 |
| Hopper-v5 | 3M | 2690 ± 806 | 2752 ± 706 | 3147 ± 613 |
| Swimmer-v5 | 1M | 75 ± 30 | 80 ± 25 | 102 ± 12 |
| Swimmer-v5 | 3M | 100 ± 38 | 98 ± 32 | 128 ± 12 |
| Walker2d-v5 | 1M | 4111 ± 485 | 4044 ± 740 | 4021 ± 1126 |
| Walker2d-v5 | 3M | 4520 ± 908 | 4638 ± 901 | 5331 ± 686 |
| PandaPush-v3 | 1M | -31.7 ± 9.3 | -28.7 ± 10.1 | -25.5 ± 9.0 |
| PandaPush-v3 | 3M | -7.6 ± 1.0 | -7.4 ± 1.1 | -10.8 ± 10.2 |
| PandaSlide-v3 | 1M | -49.8 ± 0.4 | -46.9 ± 4.7 | -47.8 ± 3.7 |
| PandaSlide-v3 | 3M | -43.1 ± 6.2 | -41.8 ± 5.8 | -43.5 ± 6.3 |
| **# of best performances** | | 3 | 5 | **13** |

underperformed, whereas our proposed LinFFs initialisation allowed a wider frequency range to perform to a certain degree. Our proposed enhancement are much less sensible to the frequency hyperparameter choice.

To understand why, we first define a tool to study the whole encoded state information $\mathcal{D}_{\text{batch}}$ directly. We define the overall Cross Energy Spectral Density (overall CSD) by summing together the CSD for every combination of state dimension pairs in the frequency domain as in Equation (10). This approach provides an overall view of state without the need to examine the cross-correlation and energy shifts of each state dimension pair individually. $j$ is the total number of state dimensions depending on the task.

$$\sum_{\substack{k,l \\ 0 \le k < l \le j}} S_{kl}(f) = \sum_{\substack{k,l \\ 0 \le k < l \le j}} \sum_{\tau=-\infty}^{\infty} R_{kl}[\tau] e^{-i2\pi f \tau} \tag{10}$$

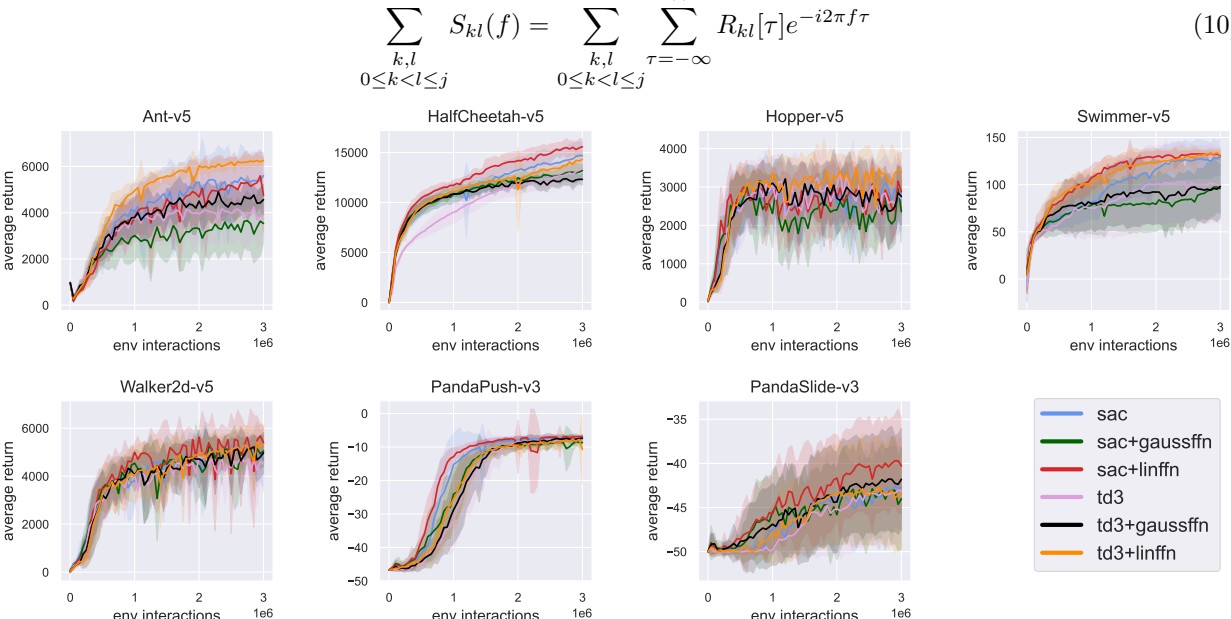

Figure 5: Results on 5 gymnasium tasks and 2 panda-gym tasks.

Using overall CSD, we can observe that with lower frequencies $f$, the energy of the whole state dimensions $\mathcal{D}_{\text{batch}}$ become more concentrated in the low frequency spectrum. A numerical example of the overall CSD is shown in Figure 6 (using the same input states as in Figure 4).

The overall energy distribution of the states without FFs encoding was generally distributed evenly across all frequencies with slightly higher energy concentrated in the low frequency spectrum, indicating that all states $s_c$ are generally changing at a similar amplitude for all frequencies within the whole state dimensions.

Table 2: Performance on the gymnasium and panda-gym benchmark with SAC as base algorithm at 1M and 3M timesteps, calculated over 10 seeds, with $\pm$ denoting standard deviation. The highest performance is highlighted in blue. Any performance which is not statistically significantly worse than the highest performance (according to a Welch's t-test with significance level 0.05) is also highlighted in yellow.

| Task | Timestep | SAC | SAC+GaussFFN | SAC+LinFFN |
|---|---|---|---|---|
| Ant-v5 | 1M | 4052 ±1147 | 3706 ±1049 | 3094 ± 821 |
| Ant-v5 | 3M | 5606 ±947 | 3315 ±942 | 5507 ±845 |
| HalfCheetah-v5 | 1M | 11034 ±936 | 11218 ±659 | 11732 ±699 |
| HalfCheetah-v5 | 3M | 14673 ±1350 | 12950 ±1226 | 15611 ±275 |
| Hopper-v5 | 1M | 2909 ±860 | 2296 ±783 | 2773 ±689 |
| Hopper-v5 | 3M | 2886 ±959 | 2318 ±879 | 2877 ±565 |
| Swimmer-v5 | 1M | 81 ±17 | 67 ±18 | 107 ±12 |
| Swimmer-v5 | 3M | 129 ±18 | 81 ±28 | 131 ±5 |
| Walker2d-v5 | 1M | 3789 ±1008 | 3937 ±672 | 3883 ±1172 |
| Walker2d-v5 | 3M | 4693 ±1255 | 4972 ±657 | 5015 ±1669 |
| PandaPush-v3 | 1M | -15.2 ±9.2 | -24.1 ±7.7 | -12.9 ±2.6 |
| PandaPush-v3 | 3M | -7.2 ±0.3 | -8.7 ±1.4 | -6.9 ±0.2 |
| PandaSlide-v3 | 1M | -47.5 ±2.8 | -45.5 ±4.8 | -45.7 ±4.2 |
| PandaSlide-v3 | 3M | -42.7 ±4.7 | -44.0 ±5.7 | -40.3 ±5.4 |
| **# of best performances** | | 6 | 4 | **14** |

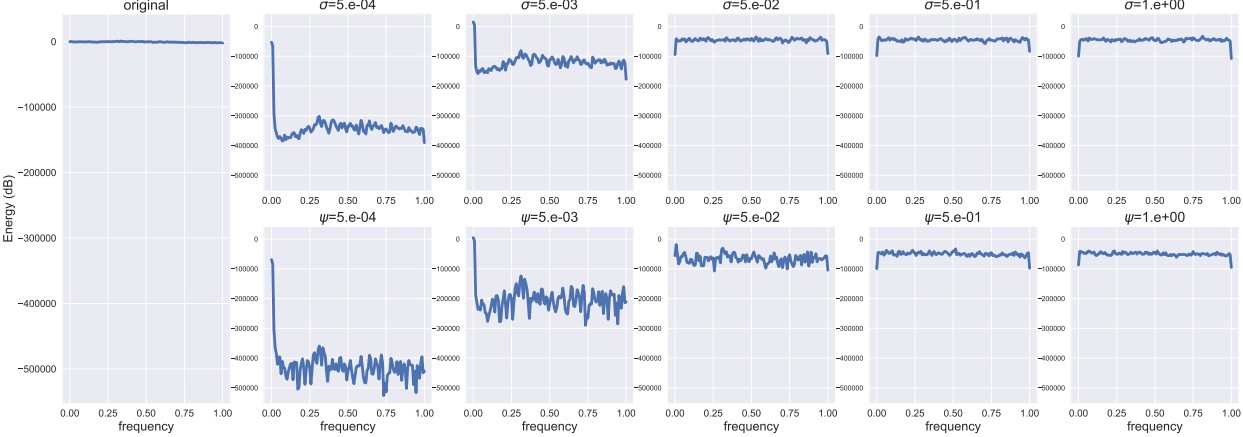

Figure 6: An example showing the overall CSD of encoding the input states $\mathcal{D}_{\text{batch}}$ with RFFs and LinFFs. The overall CSD of the pre-encoded states is the left-most sub-figure. Top row: GaussFFN. Bottom row: LinFFN. From left to right, the frequency sampling hyperparameter ($\sigma$ or $\psi$) increases. After encoding with FFs, the overall energy becomes more augmented for lower frequencies but remains spread out across the spectrum as frequencies goes higher. Our proposed LinFFN allow more energy to remain concentrated in the low frequency spectrum when $\sigma/\psi$ goes higher compared to GaussFFN. Thus, our proposed enhancement suffer less from the drop in performance when higher frequencies are used. We use the same y-axis scale for the CSD plot.

Table 3: Performance on the gymnasium benchmark with TD3 as base algorithm at 1M and 3M timesteps, calculated over 10 seeds, standard deviation is omitted for clarity. Tasks are all v5 versions. We show the performance related to the best TD3 with FFs. Performance within ±20% of the best performance is highlighted in green. With our proposed enhancement, FFs can show much less sensitivity to the frequency hyperparameter and can even achieve comparable results.

| Task | Timestep | TD3 | + GaussFNN | + GaussFFN (high-freq) | | |
|---|---|---|---|---|---|---|
| $\sigma$ | | | | 1.0 | 5.0 | 10.0 |
| Ant-v5 | 1M | 3735 | 4494 | 938 (-79%) | 934 (-79%) | 945 (-79%) |
| Ant-v5 | 3M | 3919 | 6137 | 904 (-85%) | 2346 (-62%) | 1386 (-77%) |
| HalfCheetah-v5 | 1M | 9765 | 10522 | 623 (-94%) | 646 (-94%) | 832 (-92%) |
| HalfCheetah-v5 | 3M | 12483 | 13125 | 7496 (-43%) | 5844 (-55%) | 4252 (-68%) |

| Task | Timestep | TD3 | + LinFFN | + FNN(linear) (high-freq) | | |
|---|---|---|---|---|---|---|
| $\psi$ | | | | 1.0 | 5.0 | 10.0 |
| Ant-v5 | 1M | 3735 | 4950 | 4021 (-19%) | 4235 (-14%) | 4717 (-5% ) |
| Ant-v5 | 3M | 3919 | 6260 | 6157 (-1%) | 6228 (-0%) | 5990 (-4%) |
| HalfCheetah-v5 | 1M | 9765 | 10856 | 10530 (-3%) | 8479 (-22%) | 4515 (-58%) |
| HalfCheetah-v5 | 3M | 12483 | 14257 | 13238 (-7%) | 12375 (-13%) | 6221 (-56%) |

By encoding the states with low-frequency FFs, the energy of the whole encoded state information is essentially shifted towards lower frequency spectrum regardless how the frequencies are generated. If too high frequencies were employed, the overall energy is shifted towards more toward higher frequencies spectrum, resulting a evenly distributed overall energy distribution.

Based on the empirical performance shown in Table 3, we demonstrate that the overall energy of $\mathcal{D}_{\text{batch}}$ *should* be concentrated in the low frequency spectrum. As shown in Section 4.1, more meaningful state information can be encoded at lower frequencies, and thus, the energy spectrum also highlights the advantage of using low-frequency FFs, where more meaningful information about states and dynamics can be encoded. Employing RFFs with higher frequencies disrupts this natural energy distribution. Both meaningful and less meaningful information about states and dynamics were encoded at the same energy, leading to poor performance. Our proposed LinFFs initialisation allow more energy to concentrate in the low frequency spectrum even at large frequency hyperparameter, which in turn reduces the sensitivity of the final performance.

This intuition can also be empirically verified using the illustration presented in Figure 6. We observed that states encoded with LinFFs exhibited a greater concentration of energy in the low frequency spectrum and remain so even as frequency goes higher. Where as for RFFs the energy becomes evenly distribution at high frequency spectrum, hurting performance. The cross-correlation plot for the LinFFs using the same states $s_k$ for plotting Figure 4 can be found in Section B.

Furthermore, we also study the effect of formulation of FFs with cosine & sine components (the default), with cosine component only or with sine component only. We find that generally having both components is better. This can also be empirically explained using the cross-correlation and CSD plots using only cosine or sine components. More details are shown in Section C.

# 6 Related Works

## 6.1 Fourier Features for RL

Previous research has focused on employing FFs for RL, including that of (Brellmann et al., 2023; Konidaris et al., 2011; Li & Pathak, 2021; Yang et al., 2022). In particular, (Li & Pathak, 2021; Yang et al., 2022) addressed the problem from an approximate bias perspective in both actor and critic networks using tools such as NTK (Jacot et al., 2018). (Brellmann et al., 2023) employed FFs only in the critic network and proposed a simplified calculation of FFs while also verifying performance gains with the on-policy algorithm

PPO (Schulman et al., 2017). The primary distinction of our work from previous studies lies in our unique method for generating frequencies. We also encode only the state to the policy to isolate and study the effect of changing state information caused by employing FFs.

### 6.2 Fourier Features Theories and Variants

Most theoretical work discussing the benefits of FFs in deep neural networks have been within the context of image tasks. The original work proposing the theoretical benefits of FFs in deep neural networks was introduced by (Tancik et al., 2020), where both cosine and sine FFs were used. (Sitzmann et al., 2020) explored an alternative approach by employing FFs as a periodic activation rather than as an encoding function. (Yuce et al., 2022) showed that FFs can be interpreted as a structured signal representation dictionary, where the periodic activation's non-linearity dictates the units of the dictionary. Additionally, (Saragadam et al., 2023) extended FFs to wavelets, citing research from harmonic analysis, showing that Fourier-based methods are suboptimal for typical vision tasks. These advancements have the potential to further enhance the performance of RL tasks if thoroughly adopted.

### 6.3 State Encoding beyond Fourier Features for RL

Beyond encoding features with FFs, several methods discuss encoding states in the context of RL. For example, TD7 (Fujimoto et al., 2023) jointly encodes both state $s_k$ and action $a_k$ into a state-action embedding $z_k^{sa}$, which is trained against estimating the next state $s_k'$. In addition to encoding states, (Chandak et al., 2019; Hausknecht & Stone, 2016) aimed to parametrise action spaces to learn a more compact action representation.

## 7 Conclusion and Limitations

As demonstrated in previous works (Li & Pathak, 2021; Yang et al., 2022), the use of RFFs with lower frequencies can achieve state-of-the-art performance. In this work, we delved into the disparity between employing lower frequencies in FFs in the context of RL and the higher frequencies commonly used in image tasks. We demonstrated that previous theories based on NTK could not comprehensively explain what happens to the state information and dynamics. We can also observe that randomly assigning the frequencies via a zero-mean Gaussian distribution, as has been done in previous works (Brellmann et al., 2023; Li & Pathak, 2021; Yang et al., 2022) can cause degradation in cross-correlation when higher frequencies are used.

The overall goal of this work was not to surpass the current state-of-the-art results regarding use of FFs. Instead, our goal was to present an alternative perspective on the underlying phenomenon when employing FFs. We offer an alternative perspective on how employing low frequencies in FFs can influence the cross-correlations and shift the corresponding energy distribution towards the low frequency spectrum. None of the above phenomena are observable with higher frequencies. We propose a simple enhancement to the original FFs formulation based on our observations and show that our proposed adjustments can stabilise the previous weaknesses of employing FFs for RL.

Although we showed that cross-correlation and energy shift of the state information play an important part empirically, how to incorporate these findings into further algorithms were not thoroughly discussed in the work. One potential direction for future algorithms is to consider how cross-correlation among state dimensions can be used to understand different failure modes. We also encourage further discussions on using these empirically findings and expand to discrete tasks.

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

# A   Toy example for Cross-correlation and Plotting Details for Figures in Section 4, Section B, Section C and Section E

In this section we add some intuitive visualisation on cross-correlation using some toy examples.

For the upper row, we generated a 1-D sequence $X$ from a full-ranged sine function that is linearly spaced from (0 to 20), we shift $X$ by a lag ($\tau$) of 3 plus some Gaussian noise $\mathcal{N}(0, 0.1)$ and zero-pad the front. We denote this shifted version as $Y$. We then calculate the cross-correlation between $X$ and $Y$. The x-axis of resulting cross-correlation is the time tag $\tau = [-10, 10]$. The plots are shown in Figure 7, where X is shown in the left subfigure, $Y$ is shown in the middle subfigure and the right subfigure is the cross-correlation, showing maximum lag at 3, as we would expect. Furthermore, there also more cross-correlation at other time-lags. showing that the two sequences remain similar for a longer period of time.

For the bottom row in Figure 7, we generated $X$ as a square function with values 1 at $[3, 7]$, and generated $Y$ as a shifted copy of $X$ by a lag of 5 plus plus some Gaussian noise $\mathcal{N}(0, 0.1)$. As expected, the cross-correlation $R_{xy}$ has maximum lag at $\tau = 5$, and also showing some other cross-correlation at other time-lags.

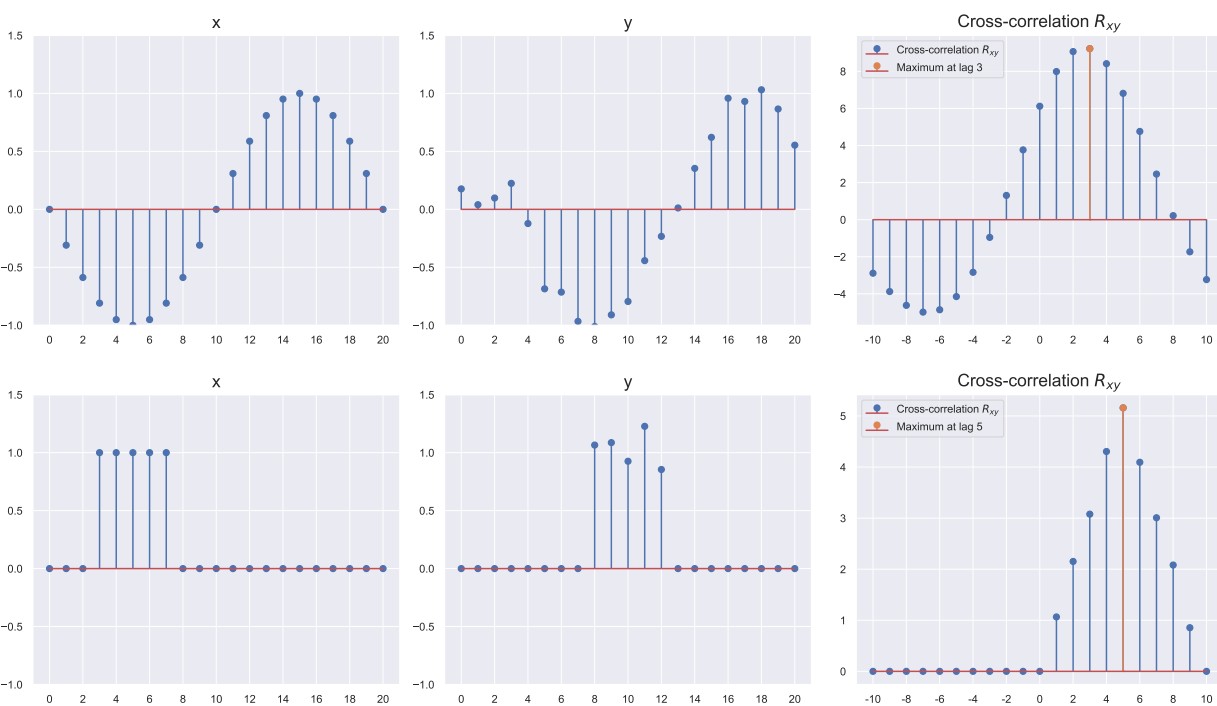

Figure 7: Toy examples of 1-D sequence $x$, $y$ and their cross-correlation.

For plots in Section 4, Section B, Section C and Section E, states are sampled from a replay buffer continuously from the task Ant-v5, with batch size 256 (timeseries length 256). For the two leftmost columns $(s_{t_k}, s_{t_l})$, they are cropped to length of 128 to improve readability. The cross-correlation column is calculated using the full 256-length timeseries $s_{t_k}$ and $s_{t_l}$. Cross-correlation is calculated via SciPy's built-in function. The CSD is calculated via matplotlib's built-in function, with no zero-padding.

# B   Figures for LinFFs

In this section, for LinFFs, we show the cross-correlation and the overall CSD (Figure 8) plots calculated using the same states as in Figure 4 and Figure 6 for the RFFs plots in Section 4. For completeness, we also show the overall CSD of LinFFs here again.

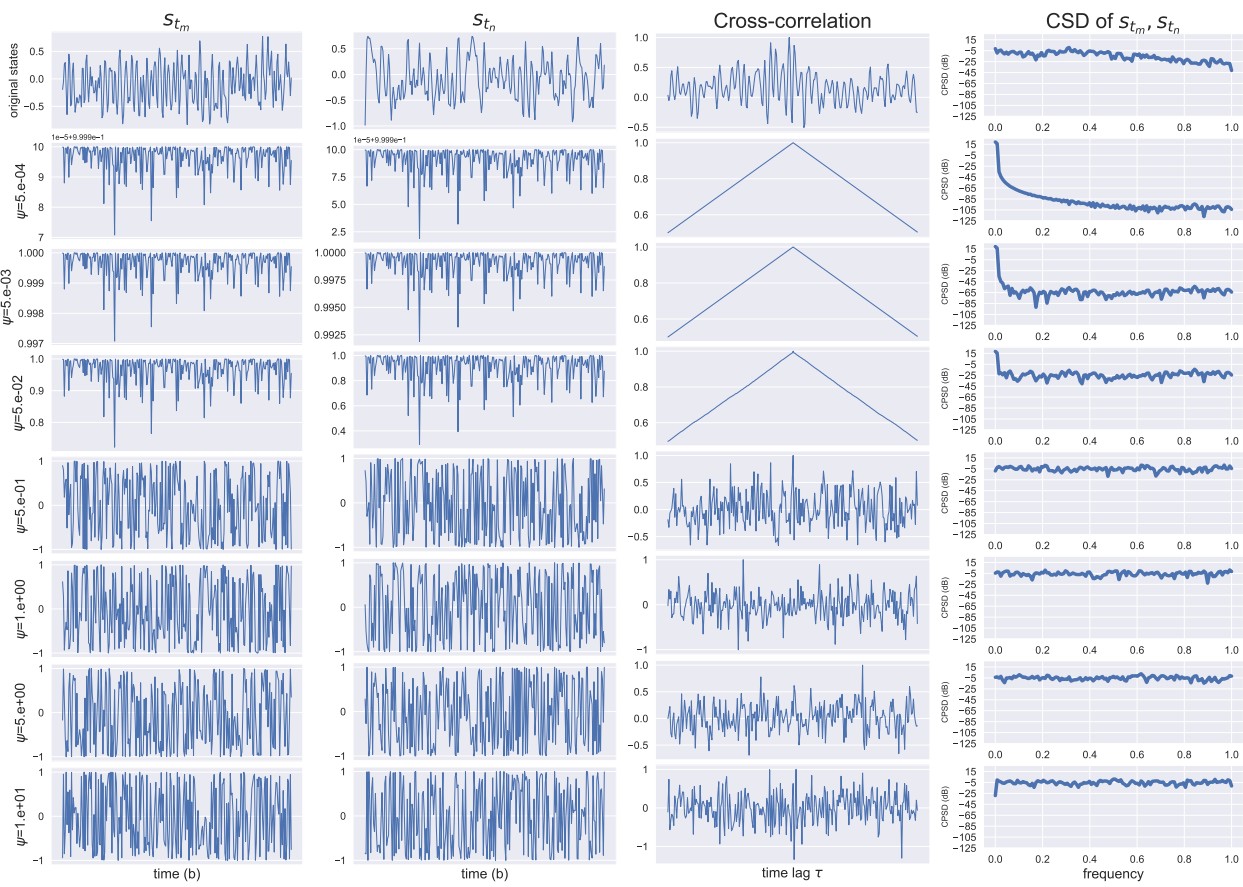

Figure 8: An example of encoding the state dimensions $(s_{t_k}, s_{t_l}) \in \mathcal{D}_{\text{batch}}$ with LinFFs. Top row denotes pre-encoded state dimensions $(s_{t_k}, s_{t_l})$. Second to last row denotes encoded states with various $\sigma$. Originally, the state dimensions are not well correlated with each other. The cross-correlation becomes more augmented for smaller $\sigma$ (middle row) but remains weak for larger $\sigma$ (third and bottom row). The right-most column denotes the CSD of the two state dimensions on the left.

## C    Effects of different formulation of FFs encodings

In the main texts, we opted to follow the original formulation of FFs, which includes both sine and cosine components. However, since cosine and sine are simply a phase shift between themselves, we can formulate FFs encoding with sine only or cosine only.

Repeating Section 2.5, the Random Fourier Features (RFFs) we used throughout the paper is defined as follows.

$$\text{RFFs}(x) = [\cos(2\pi f x), \sin(2\pi f x)]^{\mathsf{T}} \qquad \text{where frequency matrix } f \sim \mathcal{N}(0, \sigma^2) \qquad (11)$$

Depending on the way the frequencies are obtained, RFFs can also be defined with only cosine with sine function as follows. b denotes a phase shift.

$$\text{cosine-RFFs}(x) = [\cos(2\pi f x) + b]^{\mathsf{T}} \text{and} \qquad (12)$$

$$\text{sine-RFFs}(x) = [\sin(2\pi f x) + b]^{\mathsf{T}} \qquad (13)$$

$$\text{where frequency matrix } f \sim \mathcal{N}(0, \sigma^2) \text{ and } b \sim \mathcal{U}(0, 2\pi)$$

### C.1  Cross-correlation and CSD plots of different formulation of FFs encodings

Using the same timeseries states, we plot the same cross-correlation and CSD plots using cosine-FFs and sine-FFs. We can observe that generally having both cosine and sine features make the cross-correlation more pyramid-like. Having only sine features or cosine features occasionally make the FFs encoding non pyramid-like, potentially harming the performance.

### C.2  Results of different formulation of FFs encodings

We show the effects of using cosine-FFs and sine-FFs for training. Having both cosine and sine features can generally get better performance, depending on the task.

Table 4: Performance on the gymnasium benchmark with TD3 as base algorithm, varying FF formulations when FFs are used, at 1M and 3M timesteps, calculated over 10 seeds, with $\pm$ denoting standard deviation. FF formulations include cosine only, sine only and both (which is not marked).

| Task | Step | TD3 + GaussFFN | TD3 + GaussFFN(cosine) | TD3 + GaussFFN(sine) |
|---|---|---|---|---|
| Ant-v5 | 1M | $3792 \pm 810$ | $3215 \pm 1522$ | $3608 \pm 807$ |
| Ant-v5 | 3M | $4564 \pm 624$ | $3682 \pm 1301$ | $4199 \pm 1093$ |
| HalfCheetah-v5 | 1M | $10847 \pm 851$ | $10333 \pm 589$ | $10545 \pm 721$ |
| HalfCheetah-v5 | 3M | $12316 \pm 892$ | $11840 \pm 677$ | $12208 \pm 637$ |

| Task | Step | TD3 + LinFFN | TD3 + LinFFN(cosine) | TD3 + LinFFN(sine) |
|---|---|---|---|---|
| Ant-v5 | 1M | $4950 \pm 370$ | $3916 \pm 1093$ | $3797 \pm 942$ |
| Ant-v5 | 3M | $6260 \pm 190$ | $6025 \pm 460$ | $5501 \pm 965$ |
| HalfCheetah-v5 | 1M | $10856 \pm 760$ | $10424 \pm 634$ | $10576 \pm 748$ |
| HalfCheetah-v5 | 3M | $14257 \pm 1018$ | $13741 \pm 964$ | $13852 \pm 1150$ |

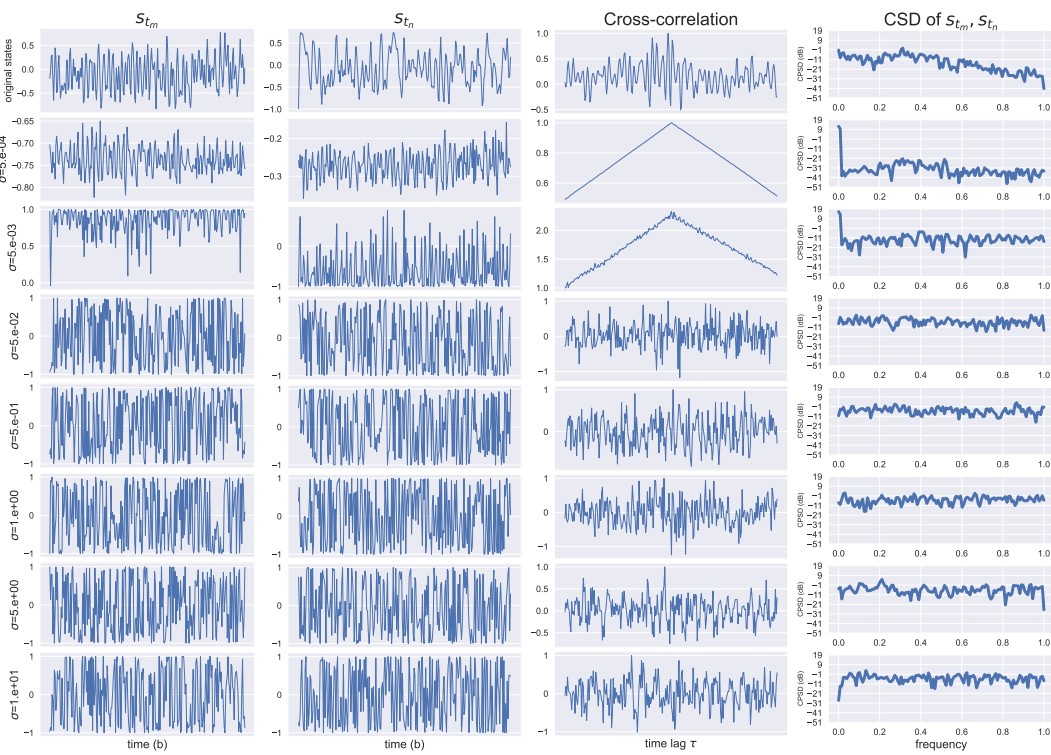

Figure 9: An example of encoding the state dimensions $(s_{t_k}, s_{t_l}) \in \mathcal{D}_{\text{batch}}$ with **cosine-RFFs**. Top row denotes pre-encoded state dimensions $(s_{t_k}, s_{t_l})$. Second to last row denotes encoded states with various $\sigma$. The right-most column denotes the CSD of the two state dimensions on the left.

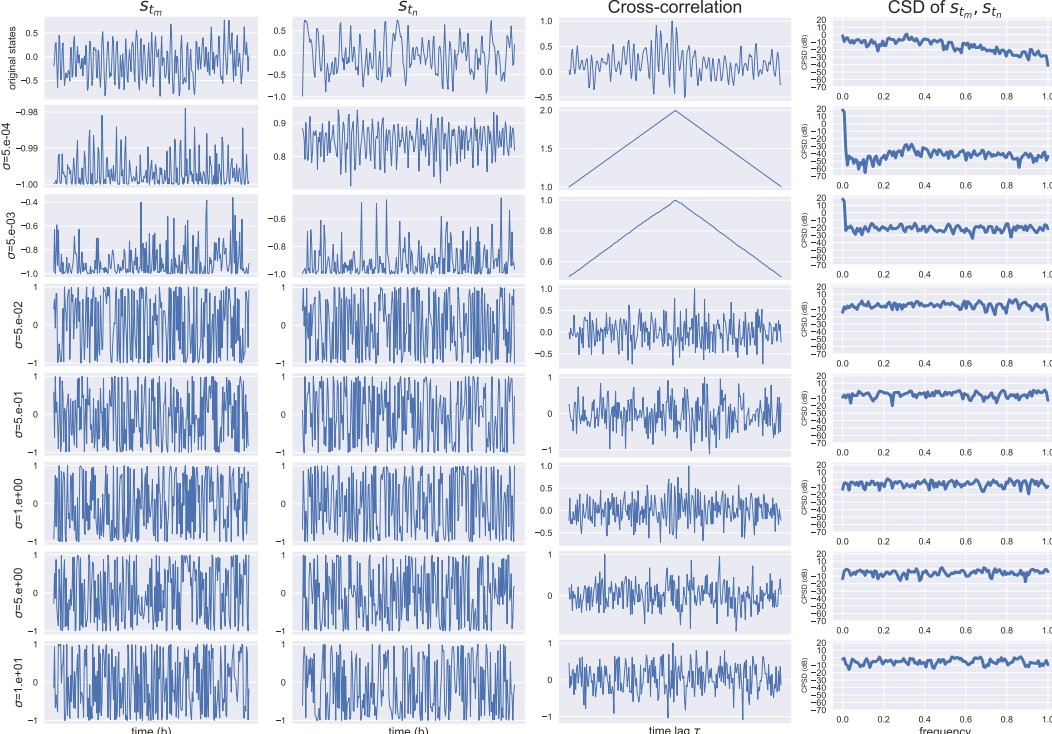

Figure 10: An example of encoding the state dimensions $(s_{t_k}, s_{t_l}) \in \mathcal{D}_{\text{batch}}$ with **sine-RFFs**. Top row denotes pre-encoded state dimensions $(s_{t_k}, s_{t_l})$. Second to last row denotes encoded states with various $\sigma$. The right-most column denotes the CSD of the two state dimensions on the left.

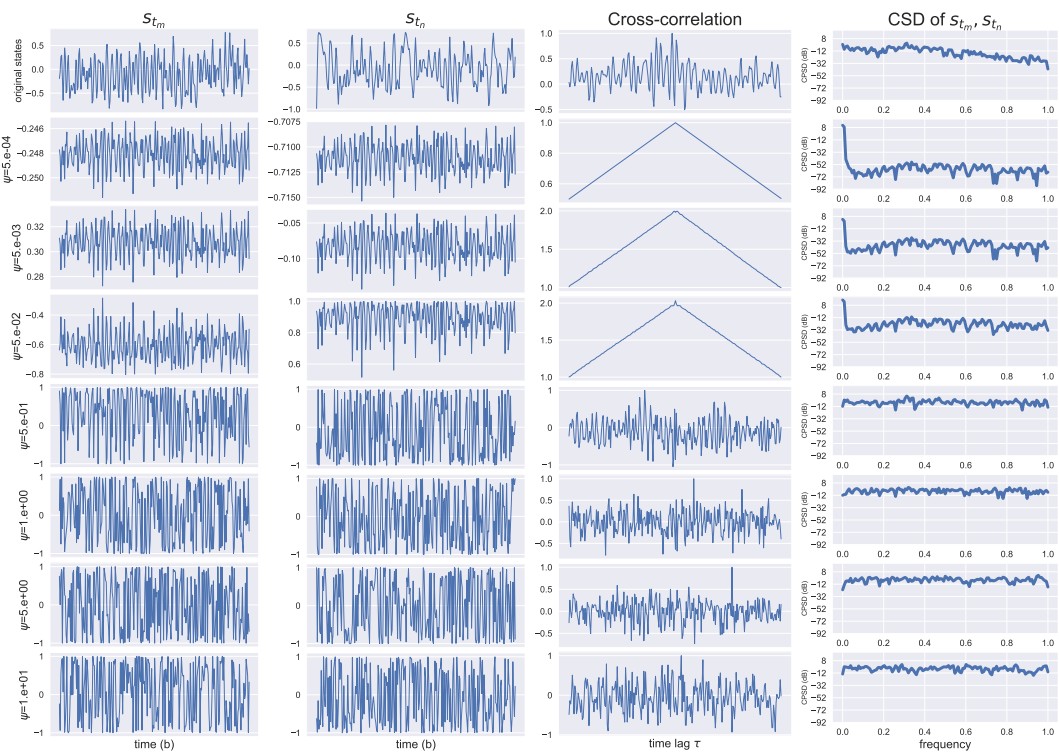

Figure 11: An example of encoding the state dimensions $(s_{t_k}, s_{t_l}) \in \mathcal{D}_{\text{batch}}$ with **LinFFN(cosine)**. Top row denotes pre-encoded state dimensions $(s_{t_k}, s_{t_l})$. Second to last row denotes encoded states with various $\sigma$. The right-most column denotes the CSD of the two state dimensions on the left.

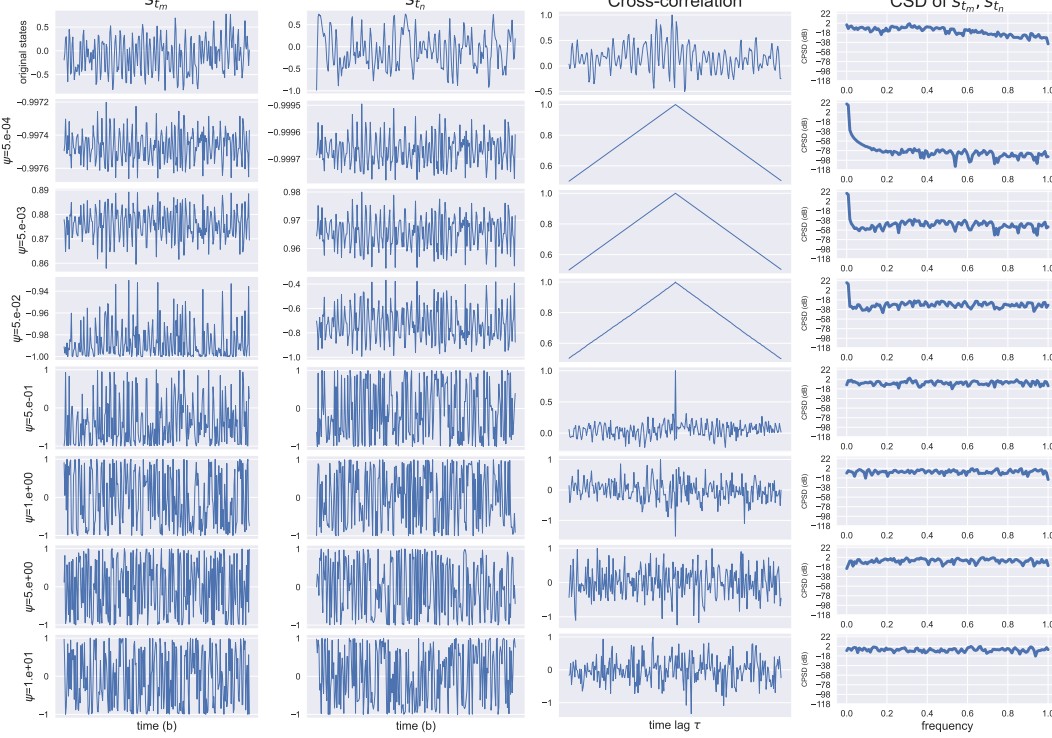

Figure 12: An example of encoding the state dimensions $(s_{t_k}, s_{t_l}) \in \mathcal{D}_{\text{batch}}$ with **LinFFN(sine)**. Top row denotes pre-encoded state dimensions $(s_{t_k}, s_{t_l})$. Second to last row denotes encoded states with various $\sigma$. The right-most column denotes the CSD of the two state dimensions on the left.

# D   Implementation details and Hyperparameters

We plan to open-source our implementation if accepted. For specific packages versions, we use gymnasium v1.0.0 (Towers et al., 2024), MuJoCo v3.2.6 (Todorov et al., 2012), and panda-gym v3.0.7 (Gallouédec et al., 2021). We feed the returned states from gymnasium package directly to the algorithms, without any modification. We implement RFFs and LinFFs as a linear layer, where the weight of the linear layer is first generated from RFFs or LinFFs. The resulting algorithm are called GaussFFN and LinFFN respectively. When FFs employ both cosine and sine features (the default), the effective $n$ for each cosine and sine are $n/2$, such that the total number of FFs would be $n$. Following previous works (Li & Pathak, 2021; Yang et al., 2022; Brellmann et al., 2023), frequencies is generated only once at the start of training. The NTK functions are calculated automatically using the Neural Tangents (Novak et al., 2020) library and uses the same configurations as in the open sourced code of (Tancik et al., 2020). The hyperparameters are listed in Table 5, and reference implementation is listed in Pseudocode 1.

Table 5: Hyperparameters for algorithms, for all tasks.

| algorithm | Parameters | Value |
|---|---|---|
| common | optimiser | AdamW (Loshchilov & Hutter, 2019) |
|  | batch size | 256 |
|  | critic learning rate | 3e-04 |
|  | actor learning rate | 3e-04 |
|  | activation function | ReLU |
|  | hidden dim | 256 |
|  | discount $\gamma$ | 0.99 |
|  | action repeat | 2 |
|  | initial random collection steps | 10000 |
|  | total train steps | $3 * 10^6$ |
|  | replay buffer size | $10^6$ |
|  | weight decay | default |
| TD3 | critic target tau $\tau$ | 0.005 |
|  | target actor noise | 0.2 |
|  | noise clipping | 0.5 |
|  | target update freq | 2 |
|  | actor update freq | 2 |
| SAC | initial temperature | 0.1 |
|  | target update freq | 1 |
|  | actor update freq | 1 |
| GaussFFN | FF dim $n$ | 256 |
|  | FF type | cosine + sine |
|  | FF trainable | True |
| LinFFN | FF dim $n$ | 256 |
|  | FF type | cosine + sine |
|  | FF trainable | True |

Table 6: Frequency related hyperparameters for individual tasks for Figure 5, Table 1 and Table 2.

| Task/Algo | LinFFN scale $\sigma$ | | GaussFFN scale $\psi$ | |
|---|---|---|---|---|
| | TD3 | SAC | TD3 | SAC |
| Ant-v5 | 0.005 | 0.0001 | 0.005 | 0.001 |
| HalfCheetah-v5 | 0.0005 | 0.001 | 0.005 | 0.001 |
| Hopper-v5 | 0.005 | 0.0005 | 0.005 | 0.0005 |
| Swimmer-v5 | 0.005 | 0.0005 | 0.005 | 0.0005 |
| Walker2d-v5 | 0.00005 | 0.005 | 0.00005 | 0.005 |
| PandaPush-v3 | 0.1 | 1.0 | 1.0 | 1.0 |
| PandaSlide-v3 | 0.1 | 0.1 | 0.1 | 0.1 |

---

**Pseudocode 1.** *Fourier Feature Networks (FFN) Details*

**Value $Q$ Network:**
▷ Identical to base algorithm, SAC or TD3.

**Value $Q$ Forward Pass:**
▷ Identical to base algorithm, SAC or TD3.

**Policy $\pi$ Network:**
▷ Initialise Fourier Features layer `l_ff` weights with either
    random scheme $f_i \sim \mathcal{N}(0, \sigma^2)$, or
    linear scheme $f_i = {}^{i\psi}/n$ for $i = 0, \ldots, n$.
▷ For other layers, identical to base algorithm, SAC or TD3.

```
l_ff = Linear(state_dim, ff_dim)
l1 = Linear(ff_dim, 256)
l2 = Linear(256, 256)
l3 = Linear(256, action_dim)  # for sac, action_dim*2
```

**Policy $\pi$ Forward Pass:**
▷ Pass the input states through `l_ff` first, without activation
▷ Here, a TD3 based algorithm is shown.

```
input = state
x = l_ff(input)
x = ReLU(l1(x))
x = ReLU(l2(x))
action = tanh(l3(x))
```

# E    Plots of Different Pairs of Cross-correlation among State Dimensions

We have shown an example of numerically calculated cross-correlations in Section 4.1. To give more examples, we sample more pairs $s_{t_j}$ in the same state $s_t$ input as in Figure 4. The goal is to show that cross-correlation between features is much more apparent when encoded with lower frequencies FFs. If we encode with higher frequencies, the cross-correlation does not manifest in an apparent way.

Specifically, we chose two set of state dimensions, one where we deem there should be a certain correlation between features before encoding and another where there are less likely certain correlation between features before encoding.

We plot the cross-correlation and CSD from $s_{t_4}$ (z-orientation of the torso (centre)) versus $s_{t_5}$ (angle between torso and first link on front left) in Figure 13 where certain correlation exists before encoding. and from $s_{t_2}$ (x-orientation of the torso (centre)) versus $s_{t_9}$ (angle between torso and first link on back left) in Figure 14, where almost no correlation exists.

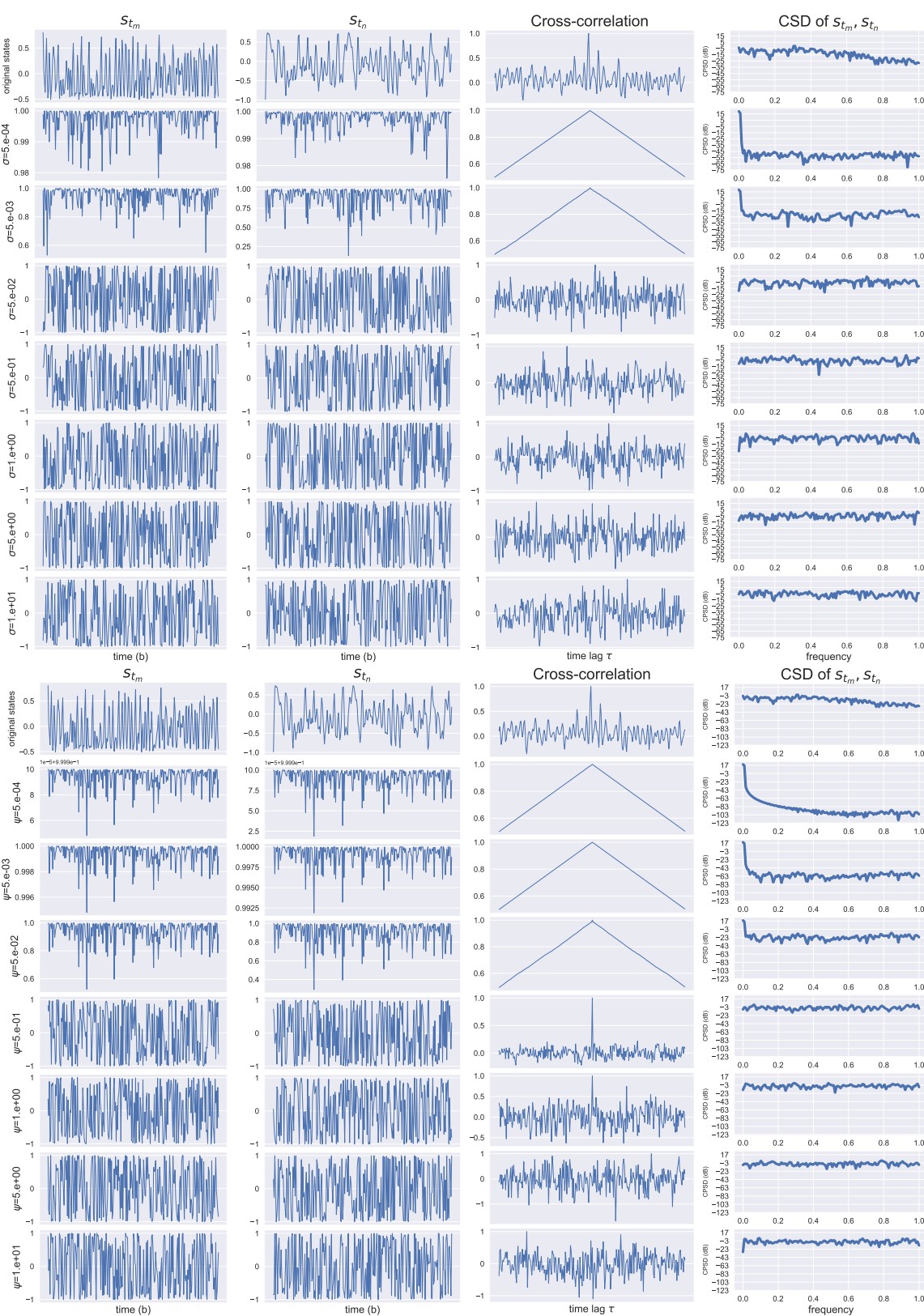

Figure 13: Cross-correlation among different pairs of state dimensions. The illustration is plotted from $s_{t_7}$ (z-orientation of the torso (centre)) versus $s_{t_{14}}$ (angle between torso and first link on front left). Upper illustration: GaussFFN. Lower illustration: LinFFN.

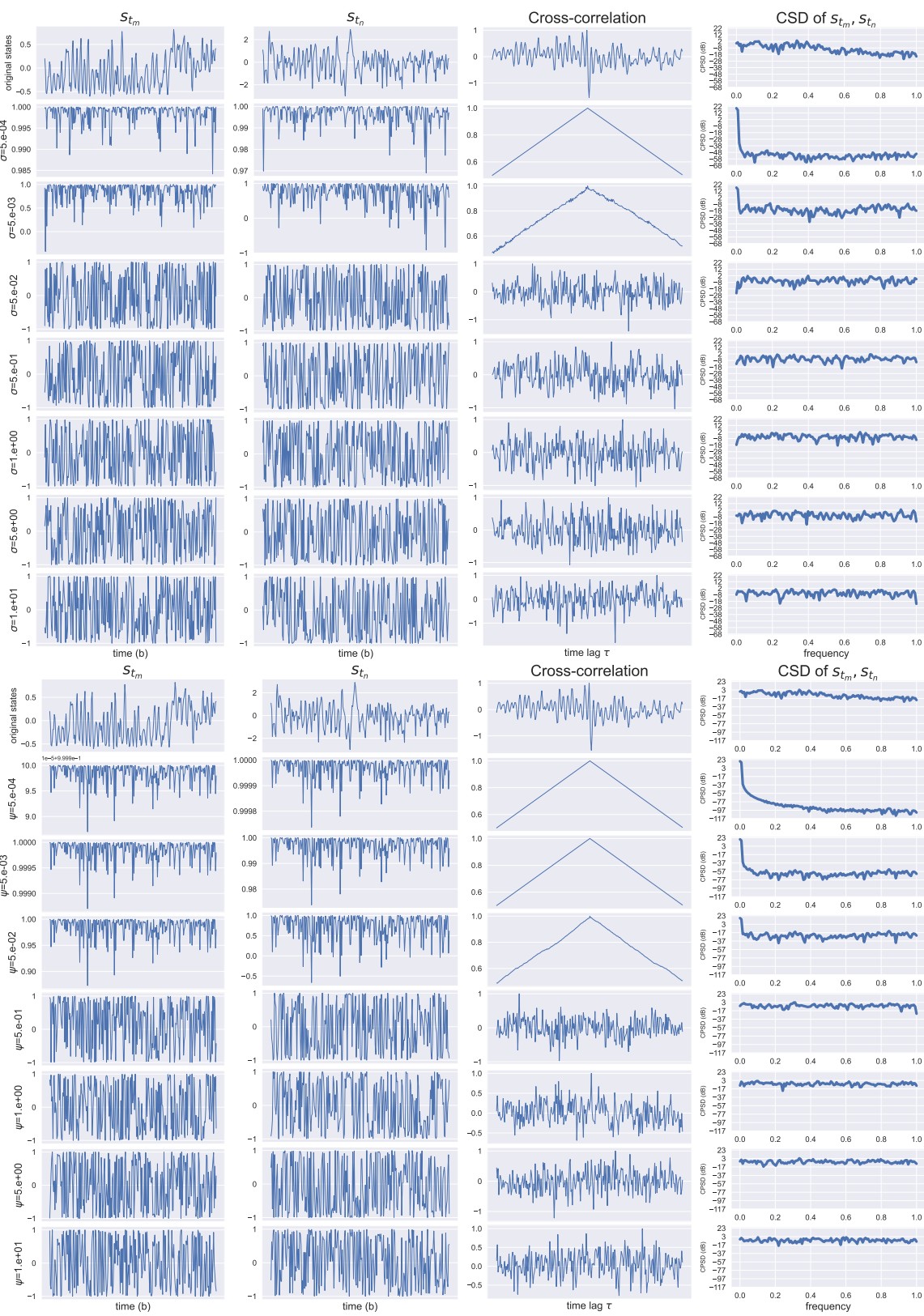

Figure 14: Cross-correlation among different pairs of state dimensions. The illustration is plotted from $s_{t_2}$ (x-orientation of the torso (centre)) versus $s_{t_9}$ (angle between torso and first link on back left). Upper illustration: GaussFFN. Lower illustration: LinFFN.

