# OpenReview forum: "On Low Frequencies Fourier Features in Reinforcement Learning"
_TMLR — Rejected by TMLR_

### Review · Reviewer_K5zx · 2025-08-07

**Summary Of Contributions:**

- The submission presents a new way of using Fourier Features in RL. Instead of using $f_i ~ \sim \mathcal{N}(0, \sigma)$ (RFFs), they propose to use $f_i = iψ/n$ called Linear Fourier Features (LinFFs).
- The authors show that lower frequencies in FFs enhance cross-correlation (true for RFFs and LinFFs).
- LinFFs can outperform the original RFFs formulation and achieve better performance on various RL tasks
- LinFFs are less sensitive to the frequency hyperparameter choice, allowing for a wider range of frequencies to perform well.

**Audience:**

Yes

**Claims And Evidence:**

No

**Requested Changes:**

1. Critical. There's a missing explanation on how FFs is used in an MLP at the end of Section 2. Please add what is the x-axis in Figure 2 and what is the formula used to obtain the parameters/inputs of the MLP from the FFs? I had to go read another paper to understand that point as it's not explained anywhere.

2. Critical. I don't understand why $\sigma$ and $\psi$ should be tested with exactly the same scale in Figure 5 and all the RL experiments. You searched $\sigma$ and $\psi$ over {1.0, 5.0, 10.0}. We need to see that if we fit $\sigma$ such that the FFs are close to those of the best $\psi$, we are indeed worst than LinFFs. Otherwise, it means I can simply reproduce LinFFs results with RFFs with another $\sigma$. Similarly for the Gaussian hypothesis, if I use a RFFs with $\mu= mean( (i {\psi}^* \times 1/n )_{i=1, ..., n} ) $ and $\sigma= mean( (i {\psi}^* \times 1/n )_{i=1, ..., n} )$, will it be worst than LinFFs?

3. "as long as they augment cross-correlation and shift energy towards low frequency spectrum in the CSD plot." why is it more important than simply increasing the final reward? Why not using the reward to measure what frequency should be used?

**Strengths And Weaknesses:**

Strengths:
- experiments on multiple environments with TD3 and SAC

Weaknesses:
- limited novelty/incremental change to RFFs
- problem on the scale of $\sigma$ and $ψ$

---

> ### Author Response · Authors · 2025-10-03
> **Response to Reviewer K5zx**
>
> Per other reviewer’ suggestions, we have renamed the practical algorithms that use trainable fourier features to reflect correctness and avoid confusion. Previously, we referred to trainable FFs built with TD3 as TD3+RFFs and TD3+LinFFs. However, RFF specifically refers to the seminal work in Rahimi & Recht [5], where frequencies are fixed (i.e. not trainable). To reflect correctness about FFs that are learned in practical algorithm, we borrow the named from Yang and al [1] and we renamed the algorithms as TD3+GaussFFN (which is the same as Yang et al.,) and TD3+GaussFFN (which uses our proposed linear spaced initialisation scheme). The same goes for the trainable FFs built with SAC, where we renamed SAC+RFFs to SAC+GaussFFN and SAC+LinFFs to SAC+LinFFN. In the updated manuscript, we have highlighted the changes from previous versions in red.
>
> ---
>
> **Q1: Critical. There's a missing explanation on how FFs is used in an MLP at the end of Section 2. Please add what is the x-axis in Figure 2 and what is the formula used to obtain the parameters/inputs of the MLP from the FFs? I had to go read another paper to understand that point as it's not explained anywhere.**
>
> Thank you for pointing this out. To clarify, the FFs are initialised in the MLP as weights of the first layer, then it is learned by typical back propagation. Furthermore, we added the code of the implementation in Appendix D to show how FFs are used. For your reference, the policy model with the learnable Fourier features, built on top of TD3, is implemented as follows:
>
> - Definition of the layers:
> ```
> l_ff = Linear (state_dim, ff_dim) # ff_dim: number of Fourier features frequencies
> l1 = Linear(ff_dim, 256)
> l2 = Linear(256, 256)
> l3 = Linear(256, action_dim)
>
> ```
> - Function to compute the action
> ```
> input = state
> x = l_ff(input)
> x = ReLU(l1(x))
> x = ReLU(l2(x))
> action = tanh(l3(x))
> ```
>
> We have updated Figure 2. What Figure 2 is plotting is the NTK function of a three-layer MLP, given inputs from [0, 1) The inputs from [0, 1] are split into M points and adding cosine/sine, we create an unit circle where the points are { \cos(2\pi j / M), \sin(2\pi j / M) }_{ j=1}^{M}, where j/M denotes the frequency. Given a fixed M, we can then calculate the NTK given the inputs varying different frequencies.  Regarding their values, we previously thought that because there is no physical meaning we opted to omit them for clarity. Nevertheless, the values of Figure 2 are, where x-axis is the distance between two points x_i and x_j. We will reflect this clarification in the related sections in the manuscript.
>
> ---
>
> **Q2: Critical. I don't understand why  and  should be tested with exactly the same scale in Figure 5 and all the RL experiments. You searched  and  over {1.0, 5.0, 10.0}**
>
> We have done some extra hyperparameter search, and the best for all FFs is shown in the updated hyperparameters table in Appendix D and also included below for your reference. Indeed, the frequency hyperparameter that results in best performance does not necessarily have to be the same for LinFFs and RFFs.
>
> | Task/Algo           | GaussFFN scale $\sigma$ | GaussFFN scale $\sigma$ 	 | LinFFN scale $\psi$ | LinFFN scale  $\psi$ |
> |-----------------------|-----------------------------|------------------------------------|--------------------------|-----------------------------|
> |                           	|        TD3             	|        SAC                 	|             TD3          	|         SAC               |
> | Ant-v5                	| 0.005                   	| 0.0001               		| 0.005                	| 0.001               	|
> | HalfCheetah-v5 	| 0.0005         	| 0.001              		| 0.005              	| 0.001             	|
> | Hopper-v5         	| 0.005              	| 0.0005              		| 0.005                	| 0.0005               	|
> | Swimmer-v5      	| 0.005              	| 0.0005              		| 0.005                	| 0.0005               	|
> | Walker2d-v5      	| 0.00005             	| 0.005               		| 0.00005             	| 0.005                	|
> | PandaPush-v3      	| 0.1            		| 1.0	               		| 1.0             		| 1.0                	|
> | PandaSlide-v3      	| 0.1             		| 0.1               		| 0.1             		| 0.1                	|

---

> > ### Author Response · Authors · 2025-10-03
> > **Continued Response**
> >
> > ---
> >
> > **Q3: "as long as they augment cross-correlation and shift energy towards low frequency spectrum in the CSD plot." why is it more important than simply increasing the final reward? Why not using the reward to measure what frequency should be used?**
> >
> > The final goal of the policy is indeed increasing the final reward. We wish to clarify that this work does not seek to re-establish why FFs does improve performance, since previous works [1][2][3] already made the argument. Rather, our work aims to address a specific gap where previous works were lacking: providing a clear justification for the crucial role of *low-frequency* components when employing FFs. In this work we argue that when using FFs, we are essentially adding cross-correlation and power to certain data in the batch and serve as a filter. We demonstrate empirically that this filtering phenomenon is uniquely prominent when encoding with lower frequencies, and cannot be readily observed when higher frequencies are used.
> >
> > ---
> >
> >
> > [1] Ge Yang et al., Overcoming the spectral bias of neural value approximation, In ICLR, 2022.
> > [2] Alexander Li et al., Functional regularization for reinforcement learning via learned fourier features, In NeurIPS, 2021.
> > [3] David Brellmann et al., Fourier features in reinforcement learning with neural networks, In TMLR, 2023.
> > [4] Quentin Gallouédec, et al., panda-gym: Open-source goal-conditioned environments for robotic learning. In NeurIPS Workshop, 2021.
> > [5] Ali Rahimi and Benjamin Recht, Random features for large-scale kernel machines, In NeurIPS, 2007
> > [6] Matthew Tancik et al., Fourier Features Let Networks Learn High Frequency Functions in Low Dimensional Domains, In NeurIPS, 2020

---

### Review · Reviewer_xUbf · 2025-08-13

**Summary Of Contributions:**

This paper investigates why Fourier Feature (FF) encodings in RL tend to work best with low frequencies. The authors show empirically that standard NTK-based explanations do not fully account for this phenomenon. They propose an alternate view based on minibatch cross-correlation and cross spectral density (CSD) of state dimensions. Accordingly, they propose LinFF: uniformly spaced, non-negative frequencies, which concentrates energy into low-frequency bands and empirically improve TD3/SAC performance on several MuJoCo tasks while reducing sensitivity to the frequency hyperparameter.

**Audience:**

Yes

**Claims And Evidence:**

No

**Requested Changes:**

- More rigourous and quantitative analysis of NTK (e.g. spectral or eigenspectral analysis)
- Benchmarking on a few more non-mujoco tasks such as Cartpole, Frozenlake or navigation.
- Ablation study to show that the performance gain comes primarily from LinFFs

**Strengths And Weaknesses:**

**Strengths**:
- The paper is well-written and organized clearly. The motivation is clear, and the results seem convincing.
- The idea of using cross-correlation and CSD as a tool to understand importance of low frequencies seems very novel, and their incorporation into mainstream RL algorithms can potentially lead to even higher gains.
- The proposed method LinFF is relatively much simpler and seems effective.

**Weaknesses**:
- The paper could benefit from a more rigorous analysis. In particular, section 3 which highlights the limitations of NTK explanation seems very qualitative at this point.
- The experiments are limited to just 4 tasks on Mujoco, and needs to be tested on more environments for the claims to be more general.
- It is unclear how much of the performance gain comes from LinFFs vs. trainable frequencies. Would be nice to have an ablation.

---

> ### Author Response · Authors · 2025-10-03
> **Response to Reviewer xUbf**
>
> Per other reviewer’ suggestions, we have renamed the practical algorithms that use trainable fourier features to reflect correctness and avoid confusion. Previously, we referred to trainable FFs built with TD3 as TD3+RFFs and TD3+LinFFs. However, RFF specifically refers to the seminal work in Rahimi & Recht [5], where frequencies are fixed (i.e. not trainable). To reflect correctness about FFs that are learned in practical algorithm, we borrow the named from Yang and al [1] and we renamed the algorithms as TD3+GaussFFN (which is the same as Yang et al.,) and TD3+GaussFFN (which uses our proposed linear spaced initialisation scheme). The same goes for the trainable FFs built with SAC, where we renamed SAC+RFFs to SAC+GaussFFN and SAC+LinFFs to SAC+LinFFN.
> In the updated manuscript, we have highlighted the changes from previous versions in red.
>
> ---
>
> **Q1 More rigourous and quantitative analysis of NTK (e.g. spectral or eigenspectral analysis)**
>
> Thank you for the feedback regarding NTK. The NTK theory states that eigenvalues of the kernel can dictate the convergence of the NN. Our NTK analysis tightly follows those of Tancik et al. [6] and Yang et al. [1]. We included the spectral plot of the NTK spectrum and also simplified the other plots in Section 3 of updated manuscript. For the spectrum plot on the right, the x-axis indicates the frequencies and the y-axis shows the corresponding eigenvalues. At low frequencies, only the lower frequencies have large eigenvalues, so they are the only ones learned through training. Increasing $\sigma$ also increases the higher frequencies’ eigenvalues, thus enabling them to learn these higher frequency characteristics faster. However, as we have stated in the paper, this is somewhat counter-intuitive because in practice we uses low frequency to learn these low frequency characteristics.
>
> ---
>
> **Q2: Benchmarking on a few more non-mujoco tasks such as Cartpole, Frozenlake or navigation.**
>
> Thank you for the feedback regarding more benchmarks. Since the target of our study is on continuous control tasks, we have done experiments on non-mujoco tasks such as LunarLanderContinous with TD3+FFN. However, since LunarLanderContinous is a relatively easy task, we find the performance differences between different FFs are not readily observable, thus omitted reporting them. Since we think the reviewer’s concern is that only tasks based on Mujoco are experimented, therefore, we also experimented on 2 more tasks from Panda-Gym[4], which are manipulation tasks and uses Pybullet as simulator. We have updated experiments section in manuscript with step plots as requested by other reviewer, and we include a table of performance at the end of training (3M) below for your reference.
>
> | **Task**       | **Timestep** | **TD3**     | **TD3+GaussFFN** | **TD3+LinFFN** |
> |:--------------:|:------------:|:-----------:|:----------------:|:--------------:|
> | PandaPush-v3   | 3M           | -7.6±1.0    | -7.4 ±1.1        | -10.8 ±10.2    |
> | PandaSlide-v3  | 3M           | -43.1 ±6.2  | -41.8 ±5.8       | -43.5 ±6.3     |
>
> | **Task**       | **Timestep** | **SAC**     | **SAC+GaussFFN** | **SAC+LinFFN** |
> |:--------------:|:------------:|:-----------:|:----------------:|:--------------:|
> | PandaPush-v3   | 3M           | -7.2 ±0.3   | -8.7±1.4         | -6.9±0.2       |
> | PandaSlide-v3  | 3M           | -42.7 ±4.7  | -44.0 ±5.7       | -40.3 ±5.4     |
>
>
> ---
>
> **Q3: Ablation study to show that the performance gain comes primarily from LinFFs**
> Thank you for the feedback. To clarify, we follow previous works [1][2][3] (works using random initialisation for FFs) and make FFs trainable (i.e. learned Fourier Features). Also citing previous works, if FFs’ frequencies are fixed after initialisation, the agent would perform very poorly to the point of being untrainable. We believe that the performance gain of our proposed method over the randomly initialised FFs arises from the different formulations of FFs, as both methods employ trainable frequencies.
>
> ---
>
>
> [1] Ge Yang et al., Overcoming the spectral bias of neural value approximation, In ICLR, 2022.
> [2] Alexander Li et al., Functional regularization for reinforcement learning via learned fourier features, In NeurIPS, 2021.
> [3] David Brellmann et al., Fourier features in reinforcement learning with neural networks, In TMLR, 2023.
> [4] Quentin Gallouédec, et al., panda-gym: Open-source goal-conditioned environments for robotic learning. In NeurIPS Workshop, 2021.
> [5] Ali Rahimi and Benjamin Recht, Random features for large-scale kernel machines, In NeurIPS, 2007
> [6] Matthew Tancik et al., Fourier Features Let Networks Learn High Frequency Functions in Low Dimensional Domains, In NeurIPS, 2020

---

### Review · Reviewer_54CH · 2025-09-26

**Summary Of Contributions:**

The paper looks at Fourier features in deep reinforcement learning as an inductive bias for feature representations. They advocate using linearly initialized frequencies over randomly sampled ones.

**Audience:**

Yes

**Claims And Evidence:**

No

**Requested Changes:**

1) Compare to prior work (learned Fourier Features) properly and clearly in the text and experiments

2) Revise analysis away from fake IID 'timeseries' and towards a correct perspective, i.e., covariance functions for state-action inputs

4) Provide step-based plots when comparing to baseline algorithms

**Strengths And Weaknesses:**

**[S1] Empirical performance gains over SAC and related work.** Tables 1 and 2 show the proposed changes improve on prior work.

**[S2] Use of statistical tests in experiments.** I appreciated the use of Welch’s t-test and 10 seeds for statistical significance.

**[W1] Repeatedly conflates random Fourier features with sinusoidal activations.**

**[W2] Clarity issues as to whether 'frequencies' are learned or fixed.**

W1 and W2 are somewhat related. Random Fourier features (Rahimi & Recht (2007)) are linear approximations to stationary kernels, using the Monte Carlo integration implicit in the inner product of the feature space to approximate the integral in Bochner's theorem to define a stationary kernel. With RFFs, the random frequencies are fixed but the lengthscale (i.e. global scaling per input dimension) can optimized in the usual kernel regression / Gaussian process fashion.
In Tanick, et al, they use RFFs on the final layer of the network, which means they are essentially learning a compositional kernel that feeds a learned neural network feature space into a squared exponential kernel approximation [A].
In contrast, SIREN (Sitzmann et al., 2020) use sinusoidal activation throughout the network.
In Yang et al., they state
> For FFN (ours), we replace the first layer in the MLP a learned Fourier features layer (LFF)

In Li et al., they also use sinusoidal activations in the first layer only (Figure 1)

In Yang et al, they also use their learned Fourier features on the first layer. Also, it's stated
> For small neural networks with limited expressivity, we found that enabling gradient updates on the RFF parameters is important for performance.

In the appendix, it's stated

> Following previous works (Li & Pathak, 2021; Yang et al., 2022; Brellmann et al., 2023), frequencies is generated only once at the start of training

However, in Table 6 it's stated that the Fourier features are trainable. If the features are learned, then figures like Figure 4 don’t make sense, because we can learn these frequencies rather than perform model selection. However, an argument could be made for careful initialization, like in the SIREN paper.

It's very strange to me to discuss Tancik et al, Li et al. and Yang et al. in a unified fashion (e.g. Section 2.6 - 3) when each paper uses 'Fourier' features in different ways. Moreover, it's not clear which method TD3+RFF refers to. If the frequencies are learned, they are not RFFs. Moreover, where they are used in the network matters.

**[W3] Repeatedly refers to relative frequencies to an unknown reference point.**

From Section 1 onwards, the text repeatedly refers to high and low frequencies for Fourier features. This is extremely vague and also depends heavily on how the data is processed. Also, the motivation for the prior work of Li et al. and Yang et al. is that sinusoidal activations overcome the spectral bias of vanilla MLPs, i.e. learn functions with higher frequency characteristics. As a result, I found it very hard to understand what insights the paper was trying to provide beyond an analysis of hyperparameters / learned parameters.

**[W4] Use of autocorrelation and cross-correlation for IID minibatched data makes no sense.**

The role of correlation measures in time series statistics is that the latent dynamics model induces temporal correlations. By randomly sampling from the replay buffer, there are no temporal correlations between datapoints; therefore, the random process is an IID one sampling from the state distribution.  If you were to model this process with a matrix normal distribution, you would end up with two covariances: the row-wise covariance in time, which is the identity due to IID sampling, and the column-wise covariance across states, which is what we really care about for feature learning and has nothing to do with timeseries!  The covariance function derived from learned feature spaces (c.f. Gaussian processes / Bayesian linear regression) computes this quantity.

**[W5] No discussion on kernel learning.**
Beyond the NTK, I believe a discussion of what kernels are and their role in constructing covariance functions of dataset would greatly improve the mathematical treatment of the paper. There is also a wealth of work on Kernel learning, e.g. [A], [B], combining non-parametric kernel methods with NN feature spaces.
I think learning a kernel / covariance function is a much better formulation of the problem, rather than the IID-replay-buffer-samples-as-a-pretend-timeseries formulation used currently.

**[W6] No step-based plots to assess convergence and sample efficiency.**
Usually deep RL papers have step-based curves to assess sample efficiency, for example Li et al. do this in Figure 5.

**[W7] Lack of discussion with Quasi Monte Carlo random Fourier features.**
To bridge the gap between random frequencies and a linear range, 'quasi' Monte Carlo methods produce low-discrepancy samples that are more uniform in their distribution. This low-discrepancy reduces the variance of the Monte Carlo approximation, which makes finite random features more 'useful' for the downstream task in a similar fashion to linearly spaced frequencies.

[A] Manifold Gaussian processes for Regression, Calandra et al.

[B] The Promises and Pitfalls of Deep Kernel Learning, Ober et al.

[C] Quasi-Monte Carlo Feature Maps for Shift-Invariant Kernels, Avron et al

---

> ### Author Response · Authors · 2025-10-03
> **Response to Reviewer 54CH**
>
> We have renamed the practical algorithms that use trainable Fourier Features to reflect correctness and avoid confusion. Previously, we referred to trainable FFs built with TD3 as TD3+RFFs and TD3+LinFFs. To reflect correctness about FFs that are learned in practical algorithm, we borrow the named from Yang and al [1] and we renamed the algorithms as TD3+GaussFFN (which is the same as Yang et al.,) and TD3+GaussFFN (which uses our proposed linear spaced initialisation scheme). The same goes for the trainable FFs built with SAC, where we renamed SAC+RFFs to SAC+GaussFFN and SAC+LinFFs to SAC+LinFFN.
> In the updated manuscript, we have highlighted the changes from previous versions in red.
>
> ---
>
> **[W3.1] refers to high and low frequencies for Fourier features.**
>
> In the field of image-processing tasks (NeRFs), the frequencies used are often in the range of (10, 100). [1][2][3] followed their analyses and based their analysis on frequencies in range of (1, 10, 100), showing the difference between NTK plots. In practice in RL, however, frequencies in the range of (0.1, 0.0005) are commonly used, and too high a range would cause RL algorithms to underperform.  In this range of (0.1, 0.0005), NTK-based analysis cannot show the whole picture of why these are needed. In this spirit, frequencies that we referred as 'low' are relative to the frequencies commonly used in image-processing tasks.
>
> ---
>
> **[W3.2] sinusoidal activations overcome the spectral bias of vanilla MLPs, i.e. learn functions with higher frequency characteristics**
>
> We are not arguing that FFs does not overcome spectral bias, but we find counter-intuitive about the way Yang et al use *low frequencies* for training RL algorithms. As you have mentioned, the goal was to learn functions with higher frequency characteristics, but in practice low frequencies are used, which shows that RL algorithms rely on features that predominantly represent low-frequency information. This is the part where we find interesting. An interpretation based on NTK would suggest that such an approach should induce significant underfitting of the state representation (i.e. force the network to learn the *extremely* low frequency characteristics). However, we argue that this view is insufficient. In our work, we argue that their primary impact is through the introduction of specific cross-correlations and a redistribution of power across the state dimensions within the data mini-batch.
>
> ---
>
> **Q1: Compare to prior work (learned Fourier Features) properly and clearly in the text and experiments**
>
> We apologise for the naming confusion. Indeed, all fourier feature networks are learned in our work. We will rename the baseline from Yang et al. [1] as TD3+GaussFFN and TD3+LinFFN when we update the manuscript. To clarify, the frequencies in fourier features, both our work and previous works, are initialised only once at the start of training, and thus we are directly comparing with prior work, where they use a random scheme in initialising and we use the linear spaced initialisation scheme.
>
> ---
>
> **Q2: Revise analysis away from fake IID 'timeseries' and towards a correct perspective, i.e., covariance functions for state-action inputs**
>
> The reason why we want to discuss “timeseries” is because we want to see how states correlate with each other (in the mini-batch). We are aware that since samples are randomly sampled from the replay buffer the trajectory is broken up, therefore, cross-correlation does not exist. However, this is where we find interesting; only using lower frequencies can we “rediscover” correlation between state features!
>
> That said, we further plot our analysis in Section 4, using non-IID timeseries. Specifically, we replotted all related figures (Figure 4, 6 in the main manuscript, Figures 7-13 in the Appendix) with states sampled from the replay buffer using consecutive indices, so that temporally consecutive states from the same trajectory were used. We still observe the same tendency; that low frequency is essential for capturing the correlation between states, while high frequencies fail to capture the correlation. Therefore, we argue that only using low frequencies can we capture correlation and power shifts between state features, and if frequencies are too high this correlation will be broken.
>
> ---
>
> **Q3: Provide step-based plots when comparing to baseline algorithms**
>
> We added step-based performance plots in Section 5, for clarity, we also removed some results at 500k from the Table 1 and 2. We also added extra experiments per requests of other reviewers on two manipulation tasks from panda-gym [4]. Please refer to the Section 5 in the updated manuscript for the plots.

---

> > ### Comment · Reviewer_54CH · 2025-11-18
> >
> > Thanks for addressing my comments.
> >
> > I remain quite confused by your comments to Q3
> > > The reason why we want to discuss “timeseries” is because we want to see how states correlate with each other (in the mini-batch). We are aware that since samples are randomly sampled from the replay buffer the trajectory is broken up, therefore, cross-correlation does not exist. However, this is where we find interesting; only using lower frequencies can we “rediscover” correlation between state features!
> >
> > If one were to compute the covariance matrix of a set of states (be it a minibatch or whole replay buffer), then
> > $$\mathbb{C}[\mathbf{s}] = \mathbb{E}[\mathbf{s}\mathbf{s}^\top] - \mathbb{E}[\mathbf{s}]\mathbb{E}[\\mathbf{s}]^\top $$
> > this computation generalized 1D cross-correlation and cross-covariance (i.e. whether you subtract the mean or not), to vector-valued random variables. So to 'calculate the cross-correlation for each state dimension pairs.' (as you state in the Figure 1 caption) you don't need to view 'mini-batch as timeseries' and apply 1D cross-correlation and autocorrelation, but just compute $\mathbb{E}[\mathbf{s}\mathbf{s}^\top]$. Am I missing something?

---

> ### Author Response · Authors · 2025-11-19
> **Authors' response**
>
> Thank you for the valuable feedback. We would like to clarify that cross-correlation and covariance matrix are different in the following sense. As you are aware, the cross-correlation examines correlation when time shifts (i.e. each sample’s relationship with another sample inside batch), while covariance matrix examines the relationship between different “features” of the samples as a whole (i.e the whole batch), with no time shifts. As shown in Eq. (1), the cross-correlation is defined as $R_{xy} = \sum x[t]y[t+\tau]$, which is different from the definition of the covariance matrix.
>
> We are not only interested in how state dimensions are similar/dissimilar (i.e. using the whole batch to calculate the covariance matrix between state dimensions, to our understanding this is what you are suggesting), but more importantly we are interested in “when” the state-action samples are correlated and when they are not, that is, for example, if each state dimensions are lagging each other. The batch-dimension, in our opinion, becomes important when applying Fourier Features. Interestingly, as we have shown empirically, cross correlation becomes significant in FFs when low frequencies are used, but not high frequencies.
>
> We hope we have addressed your concerns. If not, what further clarification or modifications could we make to improve it?

---

> > ### Comment · Reviewer_54CH · 2025-11-21
> >
> > >but more importantly we are interested in “when” the state-action samples are correlated and when they are not, that is, for example, if each state dimensions are lagging each other.
> >
> > As I understand it, you are computing this quantity for a minibatch sampled IID from the replay buffer, which means your cross-correlation statistics
> > $R_{xy}[\tau] = \mathbb{E}[X_t Y_{t+\tau}] =  \mathbb{E}[X Y]$ due to the 'timesteps' actually occurring from IID sampling, so there is no 'when' of interest on average and you're just looking at the 'stationary' cross-correlation of the state distribution in the replay buffer?

---

> > > ### Author Response · Authors · 2025-11-21
> > > **Authors' response**
> > >
> > > Even if the samples are randomly selected from the minibatch, what we compute is $\mathbb{E}[X_t Y_{t+\tau}]$ which is not the covariance because $X_t Y_{t+\tau}\neq X_t Y_{t}$ if $\tau \neq 0$, and $\mathbb{E}\_{X_t, Y_t \sim \mathcal{D}} [X_t Y_{t+\tau} | \tau] = \mathbb{E}\_{X_t, Y_t \sim \mathcal{D}}[X_t Y_t]$ does not hold in general. In only a special case, when $\tau = 0$, the cross-correlation becomes equal to the covariance (meaned or not). The cross-correlation column in Figure 4 shows the value of the cross-correlation $R[\tau]$ with $\tau = [-128, +128]$.
> > >
> > > As you pointed out, when computing the cross-correlation using samples from the replay buffer, the cross-correlation can be viewed as the cross-correlation of $X$ and $Y$ based on the stationary distribution induced by the behaviour policy.
> > > However, as $X_t$ and $Y_{t+\tau}$ are data points from different time steps, $\mathbb{E}[X_t Y_{t+\tau}]$ still captures the temporal relationship between $X$ and $Y$.
> > >
> > > In the previous revised manuscript following your first review, we show the cross-correlation in Figures 4 (and thus the CSD plot in Figure 6) computed using time-series data where trajectories are used continuously, and we still obtain the similar results as when it is computed with samples from the replay buffer. We hope that the revised manuscript better demonstrates our empirical findings. We also added schematic example that demonstrate the intuitive meaning of the cross-correlation to Appendix A.

---

> > > > ### Comment · Reviewer_54CH · 2025-11-24
> > > >
> > > > I'm not sure this discussion is being very productive, so I will try to be clearer
> > > >
> > > > 1) In reinforcement learning, the replay buffer is made of out variable-length random rollouts
> > > > $$\mathcal{D} = [(s_1^1,s_2^1,\dots), (s_1^2,s_2^2,\dots), \dots ] $$for state $s^k_h$ from rollout k and timestep h.
> > > >
> > > > 2) When a minibatch is sampled uniformly at random from the replay buffer, the samples are taken from across time and rollouts, so for example the minibatch states could be
> > > > $$\mathcal{D}_{\text{batch}} = [s_7^1,s_2^1, s_3^6, s_1^9, s_2^5,\dots]$$
> > > >
> > > > Therefore, the Section 2.2 heading 'Viewing the mini-batch as timeseries' is confusing to see, because usually the minibatch randomly shuffles the timeseries data and removes any temporal correlation from the replay buffer, which is central to any statistical analysis of timeseries data. I can now see that in the first paragraph of Section 4.1 there is a newly added sentence
> > > > > Specifically, the states are sampled from the replay buffer using consecutive indices, so that temporally consecutive states from the same trajectory were used
> > > >
> > > > Sentence helps clarifies my confusion, but passages like
> > > >
> > > > > batch index as a proxy for the time index
> > > >
> > > > which precede the previous quote are now confusing, because in what you are actually doing batch index is the same as the time index? Section 2.2 remains a source of confusion as this section has not been edited, and does not make it explicit that the minibatch sampling extracts subsequences to preserve temporal correlations.
> > > >
> > > > 3. The use of correlation and auto-correlation does not make sense for feature selection.
> > > > In the covariance function / Gaussian process literature, which I recommended studying in my initial review, the hyperparameters of the kernel function / feature space are chosen to improve the regression error on the dataset or a validation dataset. I have never seen autocorrelation between features used, because it's not clear to me how engineering greater correlation between features improves regression performance. In Figure 3, the lower noise range is motivated by inducing a triangular cross-correlation signal, with higher cross-correlation being 'better' for some reason. However, I think this result is not particularly notable from a feature learning perspective and instead is rather detrimental. For Fourier features, small frequencies mean that the sine and cosine values will be approximately 0 or 1, respectively. Indeed, in Figure 4 the left most trace for $\sigma$ in the second and third raw is approximately constant at a value of 1.0
> > > > Indeed, if I run
> > > > ```
> > > > import numpy as np
> > > > from scipy import signal
> > > > s = np.ones((256,))
> > > > cc = signal.correlate(s, s)
> > > > ```
> > > > I see the `cc` value also exhibits the triangular trace shown in Figure 4. Therefore, the cross-correlation is purely an artifact of making the features approximately constant and has nothing to do with expressivity. Based on your statement
> > > >
> > > > > Cross-correlation is calculated via SciPy’s built-in function.
> > > >
> > > > I assume Figure 4 was produced using similar code to my snippet. I could not see it anywhere in the supplementary code submission. If transforming input features to a shared constant value induces more visible cross-correlation, how can the authors motivate that this metric is beneficial for downstream regression performance?

---

> ### Author Response · Authors · 2025-11-26
>
> Thank you again for the feedback.
> Regarding Section 2.2, we have rewritten the Subsection to more clearly reflect what we are analysing in Section 4 of the manuscript. We have also updated the supplemental material to include the source code to run our analysis in Section 4.
>
> Regarding your third point, your understanding is correct. Using FFs with much lower frequencies transforms the input features into similar values, hence the cross-correlation between them becomes much more apparent and “triangular”-like.
>
> We present an alternative perspective to understand the significance of using lower frequencies in FFs, which empirically are shown to be essential in the contexts of RL. We have found that the range of frequencies used in previous works used naturally causes the cross-correlation to exhibit the aforementioned “triangular” behaviour. As you’ve pointed out, this frequency range effectively converts the input features to similar values, meaning that most states become correlated and only sufficiently abrupt changes of state features values remain distinguishable (i.e. the number of peaks in leftmost row and 2nd row from the left in Figure 4 become fewer).
>
> To the best of our knowledge, no prior studies have explicitly pointed out this phenomenon. While we do not provide a direct theoretical explanation connecting this feature transformation to the resulting performance gains, we believe that our empirical findings are valuable for the RL community in helping to better understand what happens when FFs are used in RL.

---

### Author Response · Authors · 2025-10-08
**Summary of work and rebuttal changes**

Dear reviewers,

Thank you again for your time in reviewing our work. We briefly summarise this work and the main changes we made to the manuscript below. We remain open to any questions and look forward to addressing any concerns the reviewers may have.

In this work, we examine the effectiveness of Fourier Features (FFs) in RL. We demonstrated empirically why NTK theory cannot fully explain why FFs requires low frequencies to work in RL.  We propose to examine the effectiveness of FFs through the lens of cross-correlation and cross spectral density (CSD), and showed that cross-correlation between states can only exist when low frequencies are used. We further propose that instead of using gaussian initialised learned FFs (GaussFFN), we can use a linear spaced initialisation scheme (LinFFN). We show that LinFFN, paired with TD3 and SAC, can achieve state-of-the-art performance while being more robust to large frequencies. The robustness of LinFFN to larger frequencies are also shown when examining with the overall CSD across the states.

In summary, we have updated the manuscript with the following points.
- We addressed IID samples concerns, where previously we have used IID samples (i.e. randomly sampled from the replay buffer) when examining cross-correlation and CSD between states. We have update the plots in the manuscript with samples from continuous indices from the replay buffer, and show that our analysis still holds: that cross-correlation between states only exist when low frequencies are used.
- We clarified that Fourier Features are learned following previous works, and reflect the concerns in naming the algorithms.
- We added two more benchmark tasks in panda-gym which contain manipulation tasks.
- We also added reference implementation in the appendix for completeness..

Please feel free to raise any questions or concerns. Sincerely,
Authors

---

### Author Response · Authors · 2025-10-17
**A friendly note**

Dear Reviewers,

Although the official rebuttal period has passed, we wanted to make ourselves available to clarify any points or answer any questions that may have arisen even after the rebuttal period.

Sincerely,
Authors

---

> ### Author Response · Authors · 2025-11-26
>
> Dear reviewers,
>
> Thank you again for all the constructive feedbacks. To ease comparison, we note below a list of changes that we incorporated into the manuscript compared to the initial submission and we also colour the text changes in red. In no particular order:
> - Changed the analysis in Section 4 from previously randomly-sampled indices from the replay buffer to temporally continuous indices. The analysis’ tendencies remain similar.
> - Changed the order of 2nd and 3rd paragraph in Section 4 (page 5) so it reads more naturally.
> - Added additional explanation regarding NTK plots.
> - Added illustrative example of cross-correlation in Appendix A.
> - Added two more manipulation-based tasks based on panda-gym which are based on the PyBullet engine.
> - Added reference source code to run the cross-correlation/CSD analysis in Section 4.
> - Added step-wise plots for the results in Section 5.
> - Removed table results at 500k to improve clarity.
> - Added pseudocode for training in Appendix D.
> - Renamed the baseline work from RFFs to GaussFFN and our proposed method from LinFFs to LinFFN.
> - Done extra hyperparameters search on FF scales and updated the results in Section 5. Updated hyperparameters listed in Appendix D.

---

### Decision · Action_Editor_Ycc6 · 2025-12-01

**Recommendation:** Reject

**Audience:**

Yes

**Audience Explanation:**

A better explanation for why low frequency Fourier features perform well in reinforcement learning could be interesting for some of TMLR's audience.

**Claims And Evidence:**

No

**Claims Explanation:**

Prior work that investigate the use of random fourier features in reinforcement learning explains the benefit of low frequencies based on NTK theory. Instead, the submission argues, that NTK can not explain the benefit of very low frequencies, since the kernel function would be very similar for two different low frequencies, despite these frequencies would result in noticable effect during reinforcement learning. The submission also provides an alternative hypothesis by arguing that the benefit is caused by larger cross-correlation between the timeseries of different state dimensions. The submission aims to support this hypothesis using plots that show that low frequencies result in pyramid-shape cross-correlation plots. However, as reviewer 54CH pointed out, these plots are not convincing, as the same pyramid-shaped plots would results for constant timeseries (it is a consequence of zero padding when shifting finite-length time series). In general, it seems clear that high cross-correlation on its own is not sufficient to explain good RL performance, since constant features would clearly be detrimental. The submission does hint at a potential explanation for the positive effects of low Fourier features by arguing that sufficiently large changes in the state will remain distinguishable. However, the current revision still fails to formulate this hypothesis properly, nor do the experiments based on time series cross-correlation seem suitable to support such claim.

Hence, I argue that the submission requires a major revision. The use of cross-correlation needs to be better motivated, in particular given that policy and value function operate on individual time steps. Furthermore, since high cross-correlation alone is not sufficient to obtain good features, the submission needs to more thoroughly investigate why low frequencies work well, despite mapping many different state to very similar values.

**Resubmission Of Major Revision:**

The authors may consider submitting a major revision at a later time.